# NEURAL CDEs AS CORRECTORS FOR LEARNED TIME SERIES MODELS

## ABSTRACT

Learned time-series models, whether continuous- or discrete-time, are widely used to forecast the states of a dynamical system. Such models generate multi-step forecasts either directly, by predicting the full horizon at once, or iteratively, by feeding back their own predictions at each step. In both cases, the multi-step forecasts are prone to errors. To address this, we propose a Predictor-Corrector mechanism where the Predictor is any learned time-series model and the Corrector is a neural controlled differential equation. The Predictor forecasts, and the Corrector predicts the errors of the forecasts. Adding these errors to the forecasts improves forecast performance. The proposed Corrector works with irregularly sampled time series and continuous- and discrete-time Predictors. Additionally, we introduce two regularization strategies to improve the extrapolation performance of the Corrector with accelerated training. We evaluate our Corrector with diverse Predictors, e.g., neural ordinary differential equations, Contiformer, and DLinear, on synthetic, physics simulation, and real-world forecasting datasets. The experiments demonstrate that the Predictor-Corrector mechanism consistently improves the performance compared to Predictor alone.

## 1 INTRODUCTION

Time-series datasets–both regular and irregular–are ubiquitous in nature, such as weather forecasting, human motion, etc (Guo et al., 2022a;b; Shahid & Fleming, 2024; Wang et al., 2024; Kurth et al., 2023). Learning time-series models from such datasets has applications ranging from energy demand forecasting, traffic and mobility prediction, weather prediction, anomaly detection, and decision-making in robotics (Zeng et al., 2022; Li et al., 2017; Stankeviciute et al., 2021; Xu et al., 2021; Chua et al., 2018). Several works focused on learning time-series models from data. There are at least two ways to train such models. Early studies focused on training the model to predict one step ahead (Basharat & Shah, 2009; Khansari-Zadeh & Billard, 2011). In recent years, most works focused on training the models to predict multiple steps ahead, such as Neural Ordinary Differential Equations (NODE) (Chen et al., 2019), LogTrans (Li et al., 2019), and others (Zhang et al., 2025; Lu et al., 2025). At inference time, learned time-series models are used to predict multiple steps ahead via autoregression (i.e., iterated multi-step forecasts), where the model's own predictions are fed back as an input. In doing so, the error in past predictions propagates to the future predictions, resulting in error accumulation. Generally, multi-step trained models result in more accurate forecasts than one-step trained models (Janner et al., 2021). Nonetheless, both types of models result in forecast trajectories diverging from the ground truth trajectories. There are variants of learned time-series models, such as Fedformer (Zhou et al., 2022) and others (Wu et al., 2022; Zhou et al., 2021) that forecast non-autoregressively (i.e., direct multi-step forecasts) yet suffer from performance degradation over long forecast horizons (Zeng et al., 2022).

As a motivating example, we train a NODE model with a fully connected feedforward neural network parameterizing its vector field on a synthetic time-series dataset from the FitzHugh-Nagumo, which is a two-dimensional dynamical system ($v$ and $w$) (Izhikevich & FitzHugh, 2006b). Fig. 1(a) shows the forecast performance of the trained NODE (Predictor [1]) on the w-dimension ($\hat{w}$), starting from an initial condition. The difference between the predicted ($\hat{w}$) and ground truth ($w$) trajectory, called the error trajectory ($e_w$), is shown in Fig. 1(b). If $e_w$ could be predicted, it would improve the

---

[1]Henceforth, the term Predictor will be used to refer to learned time-series models.

performance of the forecast ($\hat{w}$). To that end, we propose a Corrector based on Neural Controlled Differential Equation (Neural CDE) (Kidger et al., 2020). The key idea is that the error dynamics ($e_w$ and likewise $e_v$) of a trained Predictor are driven/controlled by the forecast trajectories ($\hat{w}$ and $\hat{v}$), which we provide as a control path to Neural CDE. The Neural CDE learns to predict the error dynamics ($\hat{e}_w$ & $\hat{e}_v$), shown in Fig. 1(b), of a trained Predictor. Adding these predicted errors to the forecasts yields improved forecasts ($\hat{w}_c$ and $\hat{v}_c$), shown in Fig. 1(a). We name our approach the Predictor-Corrector mechanism and demonstrate the practical instantiation of this framework by pairing the Corrector with diverse Predictors, including NODE (Chen et al., 2019), ContiFormer (Chen et al., 2024), and DLinear (Zeng et al., 2022) (Appendix H.2). Our contributions are the following:

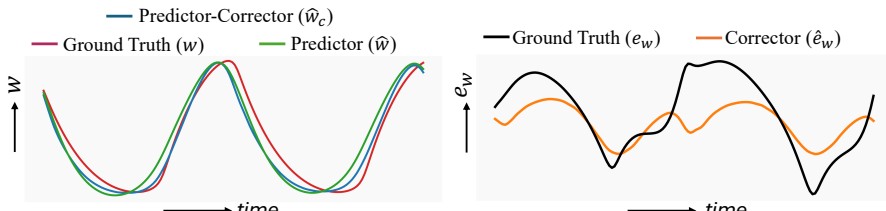

Figure 1: The performance of NODE on a synthetic dataset from FHN. **(a)** Left: The performance of Predictor with ($\hat{w}_c$) and without Corrector ($\hat{w}$) against ground truth ($w$). **(b)** Right: The Corrector's predictions of error ($\hat{e}_w$) against ground truth ($e_w$)

- We introduce a **general Predictor–Corrector framework** where the Predictor can be any learned time-series model and the Corrector is a Neural CDE. To further enhance extrapolation capability and accelerate training of our Corrector, we introduce **two effective regularization strategies**.

- Our framework is **agnostic to the form of the underlying Predictor** and compatible with different classes of forecasting models without architectural modifications. It operates seamlessly with both continuous-time (e.g., NODE, ContiFormer) and discrete-time (e.g., DLinear) models, thereby handling both regularly and irregularly sampled series.

- We show that the proposed methodology consistently improves Predictor performance across **synthetic, simulated, and real-world long-term series forecasting (LTSF) datasets**, demonstrating both robustness and broad applicability.

## 2 RELATED WORKS

**Predictor-Corrector Mechanism** We discuss three lines of work in the context of time-series modeling where the idea of Predictor-Corrector was utilized in different ways. The **first line** of work is from the forecasting domain, wherein several works introduced Correctors for forecasters (Predictors) to improve performance, though the term Predictor-Corrector was not used in those works (Chen et al., 2022; Slater et al., 2023; Liu & Meng, 2025; Zhang et al., 2022). Zhang (2003) introduced a hybrid approach in which a Multi-Layer Perceptron (MLP) was used as a Corrector to learn from the errors of the ARIMA forecasting model (Predictor). Another similar approach for drought forecasting utilized long short-term memory (LSTM) was used as a Corrector to learn the errors of the ARIMA model (Predictor) and demonstrated the superior performance of their hybrid approach over ARIMA alone (Xu et al., 2022). The **second line** of work introduced Predictor-Corrector frameworks, inspired by data assimilation approaches (Law et al., 2015). For instance, PhyDNet (Guen & Thome, 2020) is a two-branch deep architecture for video forecasting where one branch (Predictor) learns known PDE dynamics while the other branch (Corrector) learns the unknown information to correct the dynamics based on observations. KalmanNet (Revach et al., 2022) utilizes GRU (Cho et al., 2014) to fuse predictions from the Predictor and observations from the observer to learn the Kalman Gain for corrections. Such Prediction and Correction operations were extended to PDEs by Singh et al. (2024). Specifically, the authors proposed the Neural Operator with Data Assimilation (NODA) framework to correct the long-horizon predictions of neural operators based on sparse noisy measurements. The **third line** of work utilizes Predictor-Corrector methods

to better approximate the solutions to differential equations. This idea, first introduced by Diethelm et al. (2002), utilizes an explicit method for the predictor step and an implicit method for the corrector step. Li et al. (2024) extended this idea to Transformers (Vaswani et al., 2017) to introduce PCformers, a variant of the ODE transformer, for language translation. The Diffusion probabilistic models (Sohl-Dickstein et al., 2015) also benefited from this predictor-corrector paradigm, resulting in accurate and fast sampling (Zhao et al., 2023). Our contribution belongs to the first line of work, i.e., the forecasting domain, in which the Predictor-Corrector method was never formally introduced. We introduced the Predictor-Corrector mechanism to correct the Predictor's multi-step predictions in the **forecast** regime, where we don't have access to observations, in contrast to data assimilation approaches. Additionally, our proposed Corrector can correct both continuous- and discrete-time Predictors and works in regularly/irregularly sampled regimes.

**Continuous-Time Models**   RNN can update its hidden state based on an observation. However, their hidden state remains undefined between observations, making them an awkward fit for irregularly sampled time series. Several approaches used a simple exponential decay to model continuous hidden state dynamics between observations (Che et al., 2018; Rajkomar et al., 2018; Cao et al., 2018; Mei & Eisner, 2017). ODE-RNN (Rubanova et al., 2019) modeled continuous hidden dynamics between observations with a neural network akin to NODE (Chen et al., 2019). However, ODE-RNNs suffer from vanishing or exploding gradient issues, making it difficult for them to model long-term dependencies. Lechner & Hasani (2020) proposed ODE-LSTM to address these issues, effectively handling long irregularly sampled time series. Another efficient approach is a continuous-time GRU (Brouwer et al., 2019) that does a discrete update of the hidden state based on a new observation while modeling continuous dynamics between observations. While these approaches maintain continuous hidden state dynamics **between** observations, Neural CDE models continuous dynamics **across** observations. This makes Neural CDE a natural fit as a Corrector for continuous-time Predictors, where error trajectories evolve continuously over time. We also evaluate its performance on discrete-time Predictors.

# 3  PROBLEM DESCRIPTION

Consider a dynamical system whose dynamics are governed by a multivariate ordinary differential equation.

$$\dot{\mathbf{x}}(t) = \frac{d\mathbf{x}(t)}{dt} = \mathbf{f}(\mathbf{x}(t)) \tag{1}$$

where $\mathbf{x}(t) \in \mathbb{R}^D$ and $\dot{\mathbf{x}}(t) \in \mathbb{R}^D$ denote the state and first order time derivative of a $D$-dimensional system at time $t$, respectively. The $\mathbf{f}(\mathbf{x}(t))$ represents the vector-valued time derivative function. The evolving state of a dynamical system at time $t$ can be obtained by solving the ODE as:

$$\mathbf{x}(t) = \mathbf{x}_0 + \int_0^t \mathbf{f}(\mathbf{x}(\tau))d\tau \tag{2}$$

The integration starts with an initial condition $\mathbf{x}(0) = \mathbf{x}_0$ and moves time $t$ forward. We assume that the $\mathbf{f}$ is completely unknown, but it can be learnt based on observed data. Suppose we have data consisting of $N$ time series, where each time series contains $T$ irregularly sampled observations $\{\mathbf{x}_i\}_{i=0}^{T-1}$ observed at times $\{t_i\}_{i=0}^{T-1}$ with $t_0 < \ldots < t_{T-1}$ and $\mathbf{x}_i \in \mathbb{R}^D$. An approximation to the function $\mathbf{f}$, named as Predictor, can be learned via a supervised learning approach, such as NODE. The trained Predictor generates a $T$-step forecast $\{\hat{\mathbf{x}}_i\}_{i=0}^{T-1}$, either autoregressively or non-autoregressively. Let the error trajectory $\{\mathbf{e}_i\}_{i=0}^{T-1}$ denote the residual between the predicted and ground truth observations, i.e., $\{\mathbf{x}_i - \hat{\mathbf{x}}_i\}_{i=0}^{T-1}$. We hypothesize that the error dynamics can be modeled by a learned vector field $f_\theta$ driven by a control path $\mathbf{X}$. The $f_\theta$ generates a trajectory of predicted errors $\{\hat{\mathbf{e}}_i\}_{i=0}^{T-1}$ given forecasts $\{\hat{\mathbf{x}}_i\}_{i=0}^{T-1}$ as control path $\mathbf{X}$. Adding these predicted errors will yield improved forecasts, i.e., $\{\hat{\mathbf{x}}_i + \hat{\mathbf{e}}_i\}_{i=0}^{T-1}$.

## 4 METHODOLOGY

**Corrector Design.** Since the error trajectory $\{\mathbf{e}_i\}_{i=0}^{T-1}$ of a trained Predictor is driven/controlled by its forecasts $\{\hat{\mathbf{x}}_i\}_{i=0}^{T-1}$, we could model the error dynamics with a Neural CDE (Kidger et al., 2020). The Neural CDE is analogous to a continuous-time RNN whose hidden state $\mathbf{z} : [t_0, t_{T-1}] \to \mathbb{R}^C$ with $t_i \in \mathbb{R}$ is continuous in time, driven by a control path $\mathbf{X} : [t_0, t_{T-1}] \to \mathbb{R}^{D+1}$. Mathematically,

$$\mathbf{z}(t) = \mathbf{z}(t_0) + \int_{t_0}^{t} f_\theta(\mathbf{z}(s)) d\mathbf{X}(s) \ \text{ for } \ t \in (t_0, t_{T-1}], \tag{3}$$

The integral is a Riemann-Stieltjes integral. To construct the control path $\mathbf{X}(s)$, we need the forecasts from the trained Predictor for $T$ steps, i.e., $\hat{\mathbf{x}}_{0:T-1} = ((t_0, \hat{\mathbf{x}}_0), \dots, (t_{T-1}, \hat{\mathbf{x}}_{T-1}))$ with $\hat{\mathbf{x}}_i \in \mathbb{R}^D$ and $t_0 < \cdots < t_{T-1}$. The control path $\mathbf{X}(s)$ is a spline interpolant with knots at $t_0, \dots, t_{T-1}$ where $\mathbf{X}(t_i) = (\hat{\mathbf{x}}_i, t_i)$. Particularly, we use a cubic Hermite spline with backward differences (Morrill et al., 2022) (Appendix G.3). The vector field $f_\theta \colon \mathbb{R}^C \to \mathbb{R}^{C \times (D+1)}$ is a fully connected feedforward model. The variable $C$ denotes the size of the hidden state and is a hyperparameter. The initial hidden state is $\mathbf{z}(t_0) = \zeta_\phi(\hat{\mathbf{x}}_0, t_0)$, where $\zeta_\phi : \mathbb{R}^{D+1} \to \mathbb{R}^C$ is a fully connected feedforward model. The term "$f_\theta(\mathbf{z}(s)) d\mathbf{X}(s)$" is a matrix-vector multiplication. Furthermore, we employ a fully connected feedforward model as a decoder $\xi_\varphi$ to decode $\mathbf{z}(t_i)$ into a predicted error $\hat{\mathbf{e}}_i$ (Appendix G.5).

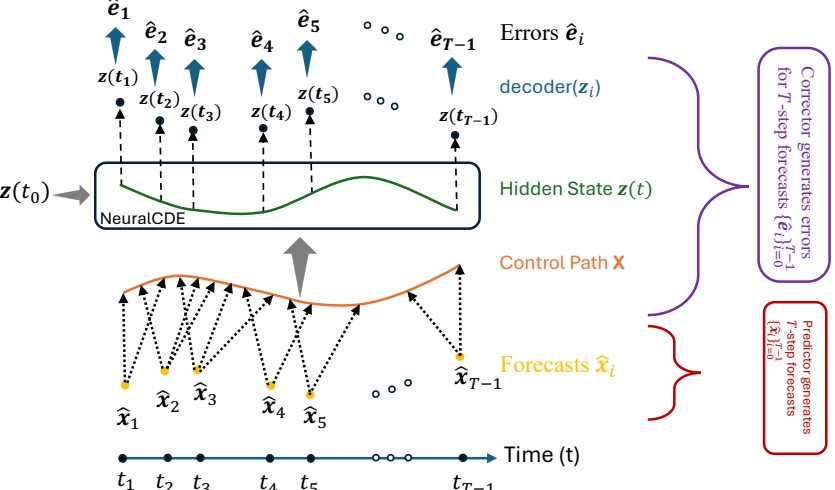

Figure 2: The proposed Predictor-Corrector methodology at inference

The training of Neural CDE as a Corrector is discussed in Algorithm 1. A trained Predictor generates $N$ trajectories of $T$-step forecasts $\{\hat{\mathbf{x}}_i\}_{i=0}^{T-1}$ and their corresponding error trajectories $\{\mathbf{e}_i\}_{i=0}^{T-1}$ at time points $\{t_i\}_{i=0}^{T-1}$. With this dataset, the parameters of $f_\theta$, $\zeta_\phi$, and $\xi_\varphi$ are learned with memory-efficient adjoint-based backpropagation with the following loss function ($\mathcal{L}$):

$$\mathcal{L} = \frac{1}{NT} \sum_{j=0}^{N-1} \sum_{i=0}^{T-1} ||\hat{\mathbf{e}}_{i,j} - \mathbf{e}_{i,j}||_2^2 \tag{4}$$

where $\hat{\mathbf{e}}_{i,j}$ denotes the predicted error at time step $i$ of trajectory $j$. We used the `diffrax` implementation of Neural CDE (Kidger, 2022). The `diffrax` offers numerical differential equation solvers in Jax (Bradbury et al., 2018). Other details regarding the training of Neural CDE are deferred to Appendix C.4.

An illustration explaining our proposed methodology at inference time is presented in Fig. 2. The Predictor generates $T$-step forecasts $\{\hat{\mathbf{x}}_i\}_{i=0}^{T-1}$ at time points $\{t_i\}_{i=0}^{T-1}$. The Corrector, a Neural CDE,

needs an initial hidden state $\mathbf{z}(t_0)$ and the continuous control path $\mathbf{X}(s)$ to generate $\mathbf{z}(t)$. The $\mathbf{X}(s)$ is built based on forecasts $(\hat{\mathbf{x}}_i)$ (Morrill et al., 2022). The continuous hidden state $\mathbf{z}(t)$ could be evaluated at observed time points $\{t_i\}_{i=1}^{T-1}$ resulting in hidden states $\{\mathbf{z}(t_i)\}_{i=1}^{T-1}$. The decoder $(\xi_\varphi)$ decodes a hidden state into a predicted error, i.e., $\hat{\mathbf{e}}_i = \xi_\varphi(\mathbf{z}(t_i))$. The addition of these errors to forecasts results in improved forecasts, i.e., $\{\hat{\mathbf{x}}_i + \hat{\mathbf{e}}_i\}_{i=1}^{T-1}$.

---

**Algorithm 1** Training the Neural CDE as a Corrector

---

**Input:** $N$ trajectories of errors $\{\mathbf{e}_i\}_{i=0}^{T-1}$ & forecasts $\{\hat{\mathbf{x}}_i\}_{i=0}^{T-1}$ from a trained Predictor. The observed times of $N$ trajectories $\{t_i\}_{i=0}^{T-1}$.
**Output:** Trained parameters $(\theta, \phi, \varphi)$ of Corrector
 1: Initialize $\theta, \phi, \varphi$
 2: **for** epoch $= 1, \dots, E$ **do**
 3:     $\mathbf{X}(s) \leftarrow$ CubicSpline$(\{(t_i, \hat{\mathbf{x}}_i)\}_{i=0}^{T-1})$     // *Control path*
 4:     $\mathbf{z}(t_0) \leftarrow \zeta_\phi(\hat{\mathbf{x}}_0, t_0)$
 5:     $\mathbf{z}(t) \leftarrow$ NeuralCDE$(\mathbf{z}(t_0), \mathbf{X}(s), f_\theta)$
 6:     // *Evaluate* $\mathbf{z}(t)$ *at* $\{t_i\}_{i=1}^{T-1}$ *to obtain* $\{\mathbf{z}(t_i)\}_{i=1}^{T-1}$
 7:     $\{\hat{\mathbf{e}}_i\}_{i=1}^{T-1} \leftarrow \{\xi_\varphi(\mathbf{z}(t_i))\}_{i=1}^{T-1}$
 8:     Minimize MSE$(\mathbf{e}_i, \hat{\mathbf{e}}_i)$     // *mean squared error*
 9: **end for**
10: **return** $\theta, \phi, \varphi$

---

### 4.1 CONTROL PATHS REGULARIZATION

We employ two regularization strategies to have the Corrector generalize beyond the time steps it was trained on, i.e., to improve its *extrapolation performance* (see 5.2).

#### 4.1.1 VARIABLE-LENGTH CONTROL PATHS $\bar{\mathbf{X}}(s)$

To construct the control path $\mathbf{X}(s)$, we need $T$-step forecast from the Predictor, i.e., $\hat{\mathbf{x}}_{0:T-1} = ((t_0, \hat{\mathbf{x}}_0), \dots, (t_{T-1}, \hat{\mathbf{x}}_{T-1}))$. During training, the last $k$ forecasts are dropped from $\hat{\mathbf{x}}_{0:T-1}$, i.e., $\hat{\mathbf{x}}_{0:T-k}$ to construct $\bar{\mathbf{X}}(s) : [t_0, t_{T-k}] \rightarrow \mathbb{R}^{D+1}$. For every forward pass, the $k$ is sampled as: $k \sim$ Uniform$\{0, 1, \dots, \eta\}$, where $\eta \in \{0, 1, \dots, T - 4\}$ is a hyperparameter. This essentially changes the limits of integration, each forward pass, in equation 3, exposing the Corrector to variable length $\bar{\mathbf{X}}(s)$, thereby improving its extrapolation performance. That is,

$$\mathbf{z}(t) = \mathbf{z}(t_0) + \int_{t_0}^{t} f_\theta(\mathbf{z}(s)) d\bar{\mathbf{X}}(s) \text{ for } t \in (t_0, t_{T-k}], \tag{5}$$

#### 4.1.2 SPARSE CONTROL PATHS $\hat{\mathbf{X}}(s)$

There are two stages of sparsity in our work for synthetic and physics simulation datasets. The first is to simulate irregularly sampled time series. It has four levels of sparsity (20%, 50%, 80%, & 100%) where each forward pass retains a corresponding percentage of points from a trajectory (Rubanova et al., 2019). The second is a regularization strategy and is designed to improve extrapolation performance. Here, an additional subset of time points is retained from an already irregularly sampled trajectory to construct a sparse control path $\hat{\mathbf{X}}(s)$. For the second stage, we introduce a hyperparameter $\kappa \in (0, 1]$ which specifies the fraction of points to retain from an irregularly sampled trajectory. This regularization reduces overfitting and improves extrapolation performance. For the LTSF dataset, we employ a $\kappa$-based regularization strategy without simulating irregularly sampled time series.

## 5 RESULTS

We evaluate our Predictor-Corrector mechanism on three categories of datasets, i.e., synthetic, physics simulation, and LTSF. NODE (Chen et al., 2022), ContiFormer (Chen et al., 2024), and

Table 1: Test MSE of NODE as a Predictor on various dynamical systems' datasets with (w/) and without (w/o) Corrector. The % ↓ shows the percentage reduction in MSE of NODE with our proposed Corrector. For both interpolation and extrapolation, reported MSE values are computed from timestep 0 up to the specified timestep ($t$) for each setting (0-$t$).

| Dynamical System | Model | Interpolation (% Observed Pts) | | | | Extrapolation (% Observed Pts) | | | |
|---|---|---|---|---|---|---|---|---|---|
| | | 20% | 50% | 80% | 100% | 20% | 50% | 80% | 100% |
| Lorenz | w/o | 2.30 | 1.061 | 1.062 | 0.887 | 12.67 | 9.68 | 8.20 | 6.46 |
| | w/ | 0.898 | 0.371 | 0.289 | 0.468 | 12.04 | 9.35 | 7.92 | 6.24 |
| | $0-t$ \| %↓ | 0–50 \| 61% | 0–50 \| 65% | 0–50 \| 73% | 0–50 \| 47% | 0–200 \| 5% | 0–160 \| 3% | 0–150 \| 3% | 0–150 \| 3% |
| Lotka Volterra | w/o | 0.101 | 0.035 | 0.027 | 0.025 | 0.153 | 0.206 | 0.040 | 0.054 |
| | w/ | 0.057 | 0.023 | 0.021 | 0.019 | 0.140 | 0.198 | 0.038 | 0.050 |
| | $0-t$ \| %↓ | 0–50 \| 44% | 0–50 \| 33% | 0–50 \| 23% | 0–50 \| 24% | 0–75 \| 9% | 0–80 \| 4% | 0–65 \| 5% | 0–70 \| 8% |
| FHN | w/o | 0.225 | 0.161 | 0.150 | 0.137 | 0.242 | 0.178 | 0.192 | 0.166 |
| | w/ | 0.164 | 0.128 | 0.093 | 0.097 | 0.231 | 0.171 | 0.183 | 0.158 |
| | $0-t$ \| %↓ | 0–50 \| 27% | 0–50 \| 20% | 0–50 \| 38% | 0–50 \| 29% | 0–75 \| 5% | 0–150 \| 4% | 0–140 \| 4% | 0–150 \| 5% |
| Glycolytic Oscillator | w/o | 0.0131 | 0.0099 | 0.0098 | 0.0097 | 0.0158 | 0.0179 | 0.0134 | 0.0160 |
| | w/ | 0.0079 | 0.0073 | 0.0072 | 0.0068 | 0.0146 | 0.0169 | 0.0127 | 0.0153 |
| | $0-t$ \| %↓ | 0–50 \| 40% | 0–50 \| 26% | 0–50 \| 27% | 0–50 \| 30% | 0–100 \| 8% | 0–110 \| 6% | 0–100 \| 5% | 0–105 \| 4% |

DLinear (Zeng et al., 2022) are used as Predictors for synthetic, physics simulation, and LTSF datasets, respectively. These experiments demonstrate that our proposed Corrector, based on Neural CDE, consistently improves the performance of diverse Predictors across a broad range of datasets.

## 5.1 BASELINES

For LTSF datasets, we improve the performance of DLinear with our Corrector and compare it with DLinear without Corrector and other transformer-based baselines. For synthetic and physics simulation datasets, we improve the performance of Predictors (i.e., NODE and ContiFormer) with our Corrector, and use the results of Predictors without Corrector as baselines. This sort of comparison is intuitive, as the primary objective of the Predictor-Corrector mechanism is to improve the performance of the Predictor.

## 5.2 INTERPOLATION & EXTRAPOLATION

For synthetic and physics simulation datasets, we evaluate the performance of Corrector to improve the Predictor's forecasts under interpolation and extrapolation settings. The interpolation evaluates the Corrector's performance within the training horizon (e.g., the first 50 time steps). We evaluate the performance of Corrector to improve the Predictor's forecasts beyond this horizon, which is called extrapolation. Specifically, extrapolation reports the maximum timestep up to which the Corrector brings at least a 3% reduction in MSE of the Predictor.

## 5.3 SYNTHETIC DATASETS

The proposed Predictor-Corrector mechanism is evaluated first on synthetic datasets generated using four multivariate ODEs, i.e., Lorenz, Lotka-Volterra, FitzHugh–Nagumo (FHN), and Glycolytic Oscillator. The closed-form of these differential equations is listed in Appendix D. The system of ODEs is solved with `solve_ivp` from `scipy` using the adaptive `RK45` solver. To produce regularly sampled trajectories, the system states are extracted at fixed time intervals ($\Delta t$). The details about the total number of trajectories, timesteps, and $\Delta t$ for each system are given in Table 10. The data was split into Train/Test with an 80/20 split; these splits were used to first train the Predictor and later the Corrector.

NODE was used as a Predictor for these datasets and trained on the first 40 timesteps. While training NODE, the irregularly sampled time series are simulated with four levels of sparsity as explained in section 4.1.2. The Corrector was trained on the first 50 time steps, which is the interpolation regime. To train the Corrector, the trajectories go through two stages of sparsity. The values of $\kappa$ and $\eta$ for

all settings are given in Table 5. The test results are reported on regularly sampled trajectories for all settings. Additional details of NODE and Corrector are given in Appendix C.1 and C.4, respectively.

Table 1 shows the MSE of NODE without Corrector (w/o) and with Corrector (w/) on the test split evaluated under interpolation and extrapolation regimes with a varying number of observed points. The Corrector improves the performance of NODE for all dynamical systems relative to the baseline NODE. The interpolation shows the reduction in MSE (%) of NODE for the first 50 timesteps as shown in row labeled ($0 - t \mid \% \downarrow$). Our Corrector (Neural CDE) is an autoregressive model and accumulates error in its predictions over time, leading to diminished correction ability at longer horizons. Therefore, we list the timesteps in the extrapolation columns for every setting up to which the Corrector brings at least a 3% reduction in MSE of NODE. For instance, when Predictor and Corrector are trained on 50% observed points for FHN, the Corrector can bring a 20% reduction in MSE from timestep 0-50 and a 4% reduction from timestep 0-150. Fig. 3 shows how the Corrector improves the Predictor within the interpolation (left of dark dotted line) and beyond it (right of dark dotted line), demonstrating correction even past the training horizon. Appendix G.9 contains long-horizon tests that show the well-boundedness of the error of the corrected forecasts.

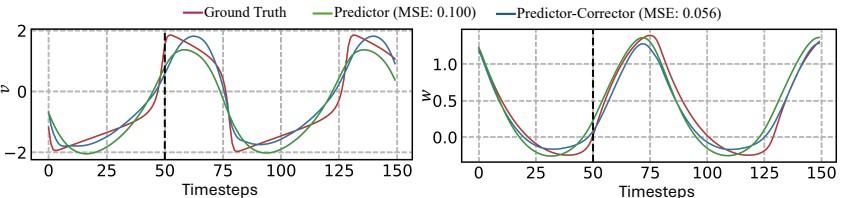

Figure 3: The performance of NODE on a test trajectory from FHN. The performance of Predictor with and without Corrector against the ground truth of **(a)** Left: $v$-dimension. **(b)** Right: $w$-dimension

## 5.4 PHYSICS SIMULATION

We evaluate the proposed Corrector on the ContiFormer, a continuous-time transformer proposed by Chen et al. (2024), as a Predictor across many simulated datasets from the MuJoCo simulator (Towers et al., 2024; Todorov et al., 2012), such as Hopper, Walker2D, Pen, and Hammer. These datasets are of varying dimensions, ranging from 11D to 46D. Each dataset has 2000 trajectories of 300 regularly sampled timesteps each. Additional details about data generation are deferred to Appendix E. The data was split into train/test with an 80/20 split. The ContiFormer was trained to predict 100 timesteps. While training ContiFormer, the irregularly sampled time series were simulated with four sparsity levels as mentioned in section 4.1.2. We use the same splits to train Corrector. The Corrector was trained on first 50 time steps, which is the interpolation regime. The test results are reported on regularly sampled trajectories for all settings. Additional details of ContiFormer and Corrector are given in Appendix C.2 and C.4, respectively. The Corrector consistently improves the performance of the ContiFormer in all settings, shown in Table 2. Within the interpolation regime, the reduction in MSE is significant for all settings. In the extrapolation columns, we report the timesteps up to which the Corrector brings at least 3% reduction in MSE of the Predictor. The Corrector shows substantial extrapolation performance for Walker2D and Pen environments. By training on 50 timesteps, the Corrector can correct the Predictor up to 200 timesteps in a few cases for Walker2D and Pen. Appendix F includes plots illustrating the performance of Predictor-Corrector on the Pen dataset.

The values of hyperparameters $\eta$ and $\kappa$, for each experimental setting in Table 2, are given in Table 6. To demonstrate the impact of Sparse Control Paths regularization on the extrapolation performance of Corrector, we investigate Walker2D with 100% of observed time points. For this setting, we chose $\kappa = 0.7$ and $\eta = 0$ as optimal values, which resulted in a 68% reduction in MSE from timesteps 0-50 and 4% reduction from 0-140, as shown in Table 2. Fig. 4(a) shows the reduction in MSE (%) of Predictor from timestep 0 up to different timesteps on the y-axis with varying values of $\kappa$ and $\eta = 0$, where the reduction in MSE (%) at timestep 50 shows the performance within the interpolation region. At $\kappa = 1.0$, where there is no regularization, the Corrector shows a substantial reduction in MSE (78%) within interpolation (0–50 timesteps) and drops below zero after timesteps 0–175. By lowering values of $\kappa$, the Corrector starts to lower its performance within the interpolation regime, with $\kappa = 0.7$ bringing only 68% reduction in MSE. However, the Corrector shows positive

Table 2: MSE of ContiFormer as a Predictor on various dynamical systems' datasets from MuJoCo (Todorov et al., 2012) with (w/) and without (w/o) Corrector. The dimensions of each dynamical system are given in brackets after its name. For interpolation and extrapolation, the reported MSE values are computed from timestep 0 up to the specified timestep ($t$) for each setting (0-$t$).

| Dynamical System | Model | Interpolation (% Observed Pts) | | | | Extrapolation (% Observed Pts) | | | |
|---|---|---|---|---|---|---|---|---|---|
| | | 20% | 50% | 80% | 100% | 20% | 50% | 80% | 100% |
| **Hopper** (11D) | w/o | 0.166 | 0.137 | 0.093 | 0.083 | 0.385 | 0.172 | 0.142 | 0.136 |
| | w/ | 0.061 | 0.018 | 0.035 | 0.028 | 0.372 | 0.149 | 0.120 | 0.132 |
| | 0–$t$ \| %↓ | 0–50 \| 63% | 0–50 \| 86% | 0–50 \| 63% | 0–50 \| 66% | 0–110 \| 4% | 0–65 \| 13% | 0–70 \| 15% | 0–60 \| 3% |
| **Walker2D** (17D) | w/o | 1.826 | 0.572 | 0.408 | 0.164 | 2.89 | 1.70 | 0.778 | 0.869 |
| | w/ | 0.721 | 0.285 | 0.149 | 0.053 | 2.68 | 1.63 | 0.750 | 0.838 |
| | 0–$t$ \| %↓ | 0–50 \| 61% | 0–50 \| 50% | 0–50 \| 63% | 0–50 \| 68% | 0–190 \| 7% | 0–180 \| 4% | 0–130 \| 4% | 0–140 \| 4% |
| **Pen** (45D) | w/o | 0.349 | 0.142 | 0.128 | 0.123 | 0.117 | 0.061 | 0.057 | 0.056 |
| | w/ | 0.150 | 0.072 | 0.050 | 0.055 | 0.110 | 0.059 | 0.054 | 0.054 |
| | 0–$t$ \| %↓ | 0–50 \| 57% | 0–50 \| 49% | 0–50 \| 61% | 0–50 \| 55% | 0–250 \| 6% | 0–190 \| 3% | 0–205 \| 4% | 0–220 \| 4% |
| **Hammer** (46D) | w/o | 0.020 | 0.0137 | 0.0106 | 0.0085 | 0.0158 | 0.0073 | 0.0049 | 0.0046 |
| | w/ | 0.012 | 0.0059 | 0.0037 | 0.0032 | 0.0150 | 0.0070 | 0.0047 | 0.0044 |
| | 0–$t$ \| %↓ | 0–50 \| 37% | 0–50 \| 56% | 0–50 \| 64% | 0–50 \| 63% | 0–100 \| 4% | 0–150 \| 4% | 0–180 \| 4% | 0–150 \| 6% |

reduction in MSE (%) within 0–200 timesteps, showing signs of improved extrapolation at the cost of little performance degradation within the interpolation region compared to $\kappa = 1.0$. Though the Corrector performance decreases within the interpolation region with the decrease in value of $\kappa$, this does not always result in an increase in extrapolation performance. Therefore, $\kappa$ should be treated as a hyperparameter. Fig. 4(b) shows the number of function evaluations (NFEs) against epochs for different values of $\kappa$ and $\eta = 0$. As the value of $\kappa$ goes down, the sparsity of a control path increases, and the NFEs show a drop, indicating reduced computational demand. At $\kappa = 0.7$, the NFEs stay close to 80 after a few initial epochs, while it is more than 90 for $\kappa = 1.0$. In summary, the sparse control paths act as a regularizer that results in improved extrapolation performance and computationally efficient training.

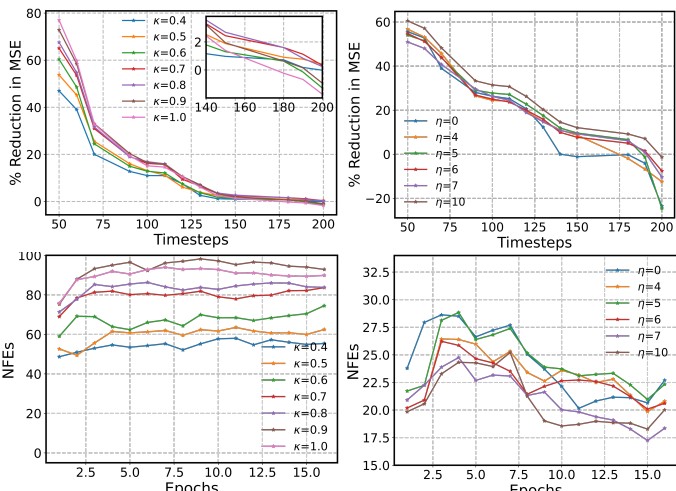

Figure 4: **(a)** Upper Left: The reduction in MSE (%) of Predictor from timestep 0 up to the timestep indicated on y-axis for different $\kappa$ values and $\eta = 0$. **(b)** Lower Left: NFEs against epochs during training for different values of $\kappa$ and $\eta = 0$. **(c)** Upper Right: The reduction in MSE (%) of Predictor from timestep 0 up to the timestep indicated on y-axis for different $\eta$ values and $\kappa = 1.0$ **(d)** Lower Right: The NFEs against epochs for different values of $\eta$ and $\kappa = 1.0$

To show the impact of the Variable Length Control Paths regularization on the extrapolation performance, we investigate Walker2D with 20% of observed points. The results reported in Table 2 for this setting are with $\kappa = 1.0$ and $\eta = 10$. Here, $\kappa = 1.0$ means that we select all of the 20% observed points each forward pass during training. Fig. 4(c) shows the reduction in MSE (%) from

Table 3: The statistics of five datasets from Table 4

| Datasets | Exchange | ETTm2 | ETTh2 | ILI | Weather |
|---|---|---|---|---|---|
| Variates | 8 | 7 | 7 | 7 | 21 |
| Timesteps | 7,588 | 69,680 | 17,420 | 966 | 52,696 |
| Granularity | 1 day | 5 min | 1 hour | 1 week | 10 min |

timestep 0 up to different timesteps indicated on the y-axis with varying values of $\eta$ and $\kappa = 1.0$. At $\eta = 0$, the Corrector generalizes very poorly and the reduction in MSE (%) drops below zero around timesteps 0–140. As we increase the value of $\eta$, the extrapolation performance improves. At $\eta = 10$, the Corrector shows a positive reduction in MSE (%) of the Predictor for timesteps 0-200. With the increase in $\eta$ value, the extrapolation performance is not guaranteed to improve. Hence, it is a hyperparameter like $\kappa$ and can be tuned. Fig. 4(d) shows the NFEs against epochs for different values of $\eta$ and $\kappa = 1.0$. The NFEs decrease as $\eta$ increases because the integrator is exposed to shorter control paths on average, resulting in faster training. This variability in the length of control paths for each forward pass improves the extrapolation performance of Corrector and leads to computationally efficient training.

We observed that the Sparse Control Paths regularization works better for densely sampled time series (50%, 80%, &, 100%), whereas it provides very little improvement in performance when the observed time points are very few, like 20%. We recommend Variable Length Control Paths regularization when the observed time points are very few. Additional results on regularization strategies include Pareto curves showing efficiency-extrapolation trade-offs under different values of $\kappa$ & $\eta$ and wall-clock training time with varying values of $\kappa$ & $\eta$ in Appendix G.6 and G.8, respectively.

## 5.5 TIME SERIES FORECASTING

We test the proposed Corrector on the challenging LTSF problems. Many transformer-based solutions were proposed for these problems, which include TimesNet (Wu et al., 2023), FEDformer (Zhou et al., 2022), Autoformer (Wu et al., 2022), Informer (Zhou et al., 2021), Pyraformer (Liu et al., 2022) and LogTrans (Li et al., 2019). Recently, DLinear (Zeng et al., 2022) was proposed to challenge the transformer-based solutions to LTSF. It is a linear decomposition-based forecasting model that outperformed various transformer-based solutions to LTSF. A few of those transformer-based solutions are mentioned earlier. We test our Corrector on the trained DLinear. There are five datasets, i.e., Exchange, ETTm2, ETTh2, ILI, and Weather. The details are given in Table 3. For ILI, there are four settings with forecast horizons $T \in \{24, 36, 48, 60\}$. The Corrector was trained on the full forecast horizon for each setting. For other datasets, there are four settings with $T \in \{96, 192, 336, 720\}$ and the Corrector was trained on the first $\{50, 100, 150, 300\}$ timesteps for each respective setting. The motivation for choosing these horizons to train the Corrector is discussed in Appendix G.1. The hyperparameters $\kappa$ and $\eta$ for each setting are given in Tables 7 and 8.

The results of the LTSF datasets are shown in Table 4. A few additional results on the LTSF datasets are given in Appendix H.1. The first column shows the results of DLinear with Corrector (w/) and the second one without Corrector (w/o). Our Corrector consistently improves the performance of DLinear across all datasets. In terms of MSE, DLinear (w/) outperforms others on Exchange, ETTm2, and Weather. On Exchange, the DLinear (w/) achieves 34% improvement over DLinear (w/o) in terms of MSE. On ETTh2 and ILI, TimesNet outperforms all others. In terms of MAE, DLinear (w/) outperforms on Exchange and Weather. Though we compare the results of DLinear (w/) against state-of-the-art transformer-based solutions, the goal of these experiments was two-fold. First, this shows that the performance of a Predictor (e.g., DLinear) for LTSF problems can be improved with our Corrector. Second, the testing of our Corrector on discrete-time Predictor (e.g., DLinear) for LTSF datasets shows its efficacy across a wide range of dynamical systems.

## 6 CONCLUSION

We proposed a Predictor-Corrector mechanism to improve the performance of learned time-series models. The Predictor is any learned time-series model and the Corrector utilizes the neural con-

Table 4: The multivariate long-term forecasting errors for five LTSF datasets (MSE/MAE; lower is better). For each dataset, the results are averaged across four forecast horizons. Best values are highlighted in bold.

| Dataset | DLinear (w/) | | DLinear (w/o) | | TimesNet | | FEDformer | | Autoformer | | Informer | | Pyraformer | | LogTrans | |
|---|---|---|---|---|---|---|---|---|---|---|---|---|---|---|---|---|
| Metric | MSE | MAE | MSE | MAE | MSE | MAE | MSE | MAE | MSE | MAE | MSE | MAE | MSE | MAE | MSE | MAE |
| Exchange | **0.242** | **0.355** | 0.369 | 0.418 | 0.416 | 0.443 | 0.518 | 0.500 | 0.613 | 0.539 | 1.550 | 0.998 | 1.485 | 1.159 | 1.402 | 0.968 |
| ETTm2 | **0.271** | 0.338 | 0.283 | 0.345 | 0.291 | **0.333** | 0.304 | 0.349 | 0.327 | 0.371 | 1.410 | 0.810 | 1.498 | 0.869 | 1.535 | 0.900 |
| ETTh2 | 0.458 | 0.457 | 0.469 | 0.464 | **0.414** | **0.427** | 0.433 | 0.447 | 0.453 | 0.462 | 4.431 | 1.729 | 0.826 | 0.703 | 2.686 | 1.494 |
| ILI | 2.35 | 1.08 | 2.40 | 1.10 | **2.139** | **0.931** | 2.846 | 1.144 | 3.006 | 1.161 | 5.136 | 1.544 | 6.007 | 2.050 | 4.839 | 1.485 |
| Weather | **0.238** | **0.286** | 0.247 | 0.300 | 0.259 | 0.287 | 0.309 | 0.360 | 0.338 | 0.382 | 0.634 | 0.548 | 0.815 | 0.717 | 0.696 | 0.601 |

trolled differential equations to learn the error dynamics of forecast trajectories of a trained Predictor. The Corrector is agnostic to the form of the Predictor and works for irregularly sampled time series as well. In addition, we propose two regularization strategies that improve the extrapolation performance of the Corrector and accelerate the training time by reducing the NFEs. The Corrector improves the performance of continuous-time Predictors like NODE and Contiformer on synthetic and physics simulation datasets, respectively. Finally, we demonstrate its efficiency on discrete-time Predictors. To that end, the Corrector is evaluated on DLinear to improve its performance, demonstrating consistent performance improvement on various real-world long-term series forecasting datasets compared to DLinear alone.

## 7 REPRODUCIBILITY STATEMENT

We have taken several steps to ensure the reproducibility of our work. The appendix provides detailed descriptions of the experimental setup, assumptions made in our work, the data generation process, and additional results. For reproducibility, we provide the relevant code as supplementary materials; upon acceptance, this will be made publicly available together with the synthetic datasets (for which we currently provide detailed generation procedures). In addition, we clearly reference all publicly available datasets used in our experiments. These resources are intended not only to verify our findings but also to support practitioners and future researchers in extending this work.

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

## A   VARIABLE LENGTH ($\eta$) AND SPARSE ($\kappa$) CONTROL PATHS HYPERPARAMETERS

We discussed two regularization methods in section 4.1 and used those for synthetic, physics simulation, and forecasting datasets. This section contains the values of the hyperparameters for both regularization for all datasets. Table 5 shows $\kappa$ & $\eta$ for synthetic datasets. Only Sparse Control Paths regularization was used for all settings. Table 6 shows $\kappa$ & $\eta$ for physics simulation datasets. As discussed in the section 5.4, we used Variable Length Control Paths ($\eta$) regularization for the settings when the observed points are very few, like 20%, or when Sparse Control Paths ($\kappa$) were not enough to provide generalization, e.g., Hopper.

Table 5: Hyperparameters $\kappa$ and $\eta$ for synthetic datasets in Table 1

| % Observed Pts | 20% | | 50% | | 80% | | 100% | |
|---|---|---|---|---|---|---|---|---|
| Hyperparameters | $\kappa$ | $\eta$ | $\kappa$ | $\eta$ | $\kappa$ | $\eta$ | $\kappa$ | $\eta$ |
| Lorenz | 1.0 | 0 | 0.8 | 0 | 1.0 | 0 | 1.0 | 0 |
| Lotka Volterra | 1.0 | 0 | 0.5 | 0 | 1.0 | 0 | 0.6 | 0 |
| FHN | 1.0 | 0 | 0.4 | 0 | 0.7 | 0 | 0.6 | 0 |
| Glycolytic Oscillator | 1.0 | 0 | 1.0 | 0 | 0.5 | 0 | 0.5 | 0 |

Table 6: Hyperparameters $\kappa$ and $\eta$ for MuJoCo datasets in Table 2

| Dynamical System | 20% | | 50% | | 80% | | 100% | |
|---|---|---|---|---|---|---|---|---|
| | $\kappa$ | $\eta$ | $\kappa$ | $\eta$ | $\kappa$ | $\eta$ | $\kappa$ | $\eta$ |
| Hopper | 1.0 | 10 | 1.0 | 10 | 0.2 | 10 | 0.2 | 10 |
| Walker2D | 1.0 | 10 | 0.6 | 0 | 0.6 | 0 | 0.7 | 0 |
| Pen | 1.0 | 0 | 0.6 | 0 | 0.6 | 0 | 0.8 | 0 |
| Hammer | 1.0 | 15 | 0.6 | 0 | 0.6 | 0 | 0.7 | 0 |

For long-term series forecasting (LTSF) problems, the hyperparameters are shown in Table 7 & 8. For most cases, we keep the values of $\kappa = 0.7$ & $\eta = 10$ fixed except for Weather.

Table 7: Hyperparameters $\kappa$ and $\eta$ for LTSF datasets in Table 4

| Forecast Horizon ($T$) | 96 | | 192 | | 336 | | 720 | |
|---|---|---|---|---|---|---|---|---|
| Hyperparameters | $\kappa$ | $\eta$ | $\kappa$ | $\eta$ | $\kappa$ | $\eta$ | $\kappa$ | $\eta$ |
| Exchange | 0.7 | 10 | 0.7 | 10 | 0.7 | 10 | 0.7 | 10 |
| ETTm2 | 0.7 | 10 | 0.7 | 10 | 0.7 | 10 | 0.7 | 10 |
| ETTh2 | 0.7 | 10 | 0.7 | 10 | 0.7 | 10 | 0.7 | 10 |
| Weather | 1.0 | 0 | 0.7 | 50 | 1.0 | 50 | 1.0 | 0 |

Table 8: Hyperparameters $\kappa$ and $\eta$ for LTSF datasets in Table 4

| Forecast Horizon ($T$) | 24 | | 36 | | 48 | | 60 | |
|---|---|---|---|---|---|---|---|---|
| Hyperparameters | $\kappa$ | $\eta$ | $\kappa$ | $\eta$ | $\kappa$ | $\eta$ | $\kappa$ | $\eta$ |
| ILI | 0.7 | 10 | 0.7 | 10 | 0.7 | 10 | 0.7 | 10 |

## B   TIME SERIES FORECASTING

Due to the limited space in the main text, we present the LTSF results averaged over four forecasting horizons $T$ in Table 4. The results for all forecast horizons are listed in Table 9. The DLinear with

Corrector (w/) consistently improved performance across datasets for each forecast horizon over DLinear without Corrector (w/o) except one case, i.e., Weather (forecast horizon 720), where there is a slight increase in MSE/MAE.

Table 9: Multivariate long-term forecasting errors (MSE/MAE; lower is better). Best value in each row is highlighted in bold.

| Methods | | DLinear (w/) | | DLinear (w/o) | | TimesNet | | FEDformer | | Autoformer | | Informer | | Pyraformer | | LogTrans | |
|---|---|---|---|---|---|---|---|---|---|---|---|---|---|---|---|---|---|---|
| Metric | | MSE | MAE | MSE | MAE | MSE | MAE | MSE | MAE | MSE | MAE | MSE | MAE | MSE | MAE | MSE | MAE |
| Exchange | 96 | **0.083** | **0.207** | 0.085 | 0.210 | 0.107 | 0.234 | 0.148 | 0.278 | 0.197 | 0.323 | 0.847 | 0.752 | 0.376 | 1.105 | 0.968 | 0.812 |
| | 192 | **0.159** | **0.295** | 0.162 | 0.297 | 0.226 | 0.344 | 0.271 | 0.380 | 0.300 | 0.369 | 1.204 | 0.895 | 1.748 | 1.151 | 1.040 | 0.851 |
| | 336 | **0.243** | **0.370** | 0.333 | 0.442 | 0.367 | 0.448 | 0.460 | 0.500 | 0.509 | 0.524 | 1.672 | 1.036 | 1.874 | 1.172 | 1.659 | 1.081 |
| | 720 | **0.482** | **0.548** | 0.896 | 0.724 | 0.964 | 0.746 | 1.195 | 0.841 | 1.447 | 0.941 | 2.478 | 1.310 | 1.943 | 1.206 | 1.941 | 1.127 |
| | Avg. | **0.242** | **0.355** | 0.369 | 0.418 | 0.416 | 0.443 | 0.518 | 0.500 | 0.613 | 0.539 | 1.550 | 0.998 | 1.485 | 1.159 | 1.402 | 0.968 |
| ETTm2 | 96 | **0.171** | **0.266** | 0.173 | 0.268 | 0.187 | 0.267 | 0.203 | 0.287 | 0.255 | 0.339 | 0.365 | 0.453 | 0.435 | 0.507 | 0.768 | 0.642 |
| | 192 | **0.235** | 0.311 | 0.239 | 0.315 | 0.249 | **0.309** | 0.269 | 0.328 | 0.281 | 0.340 | 0.533 | 0.563 | 0.730 | 0.673 | 0.989 | 0.757 |
| | 336 | **0.288** | 0.354 | 0.295 | 0.359 | 0.321 | **0.351** | 0.325 | 0.366 | 0.339 | 0.372 | 1.363 | 0.887 | 1.201 | 0.845 | 1.334 | 0.872 |
| | 720 | **0.389** | 0.422 | 0.426 | 0.439 | 0.408 | **0.403** | 0.421 | 0.415 | 0.433 | 0.432 | 3.379 | 1.338 | 3.625 | 1.451 | 3.048 | 1.328 |
| | Avg. | **0.271** | 0.338 | 0.283 | 0.345 | 0.291 | **0.333** | 0.304 | 0.349 | 0.327 | 0.371 | 1.410 | 0.810 | 1.498 | 0.869 | 1.535 | 0.900 |
| ETTh2 | 96 | **0.290** | **0.353** | 0.292 | 0.354 | 0.340 | 0.374 | 0.346 | 0.388 | 0.358 | 0.397 | 3.755 | 1.525 | 0.645 | 0.597 | 2.116 | 1.197 |
| | 192 | **0.380** | 0.418 | 0.388 | 0.422 | 0.402 | **0.414** | 0.429 | 0.439 | 0.456 | 0.452 | 5.602 | 1.931 | 0.788 | 0.683 | 4.315 | 1.635 |
| | 336 | 0.458 | 0.470 | 0.466 | 0.475 | 0.452 | **0.452** | 0.496 | 0.487 | 0.482 | 0.486 | 4.721 | 1.835 | 0.907 | 0.747 | 1.124 | 1.604 |
| | 720 | 0.705 | 0.592 | 0.729 | 0.604 | **0.462** | **0.468** | 0.463 | 0.474 | 0.515 | 0.511 | 3.647 | 1.625 | 0.963 | 0.783 | 3.188 | 1.540 |
| | Avg. | 0.458 | 0.458 | 0.469 | 0.464 | **0.414** | **0.427** | 0.433 | 0.447 | 0.453 | 0.462 | 4.431 | 1.729 | 0.826 | 0.703 | 2.686 | 1.494 |
| ILI | 24 | **2.30** | 1.07 | 2.36 | 1.09 | 2.317 | **0.934** | 3.228 | 1.260 | 3.483 | 1.287 | 5.764 | 1.677 | 1.420 | 2.012 | 4.480 | 1.444 |
| | 36 | 2.25 | 1.04 | 2.28 | 1.07 | **1.972** | **0.920** | 2.679 | 1.080 | 3.103 | 1.148 | 4.755 | 1.467 | 7.394 | 2.031 | 4.799 | 1.467 |
| | 48 | 2.25 | 1.06 | 2.35 | 1.09 | **2.238** | **0.940** | 2.622 | 1.078 | 2.669 | 1.085 | 4.763 | 1.469 | 7.551 | 2.057 | 4.800 | 1.468 |
| | 60 | 2.59 | 1.15 | 2.61 | 1.17 | **2.027** | **0.928** | 2.857 | 1.157 | 2.770 | 1.125 | 5.264 | 1.564 | 7.662 | 2.100 | 5.278 | 1.560 |
| | Avg. | 2.35 | 1.08 | 2.40 | 1.10 | **2.139** | **0.931** | 2.846 | 1.144 | 3.006 | 1.161 | 5.136 | 1.544 | 6.007 | 2.050 | 4.839 | 1.485 |
| Weather | 96 | **0.159** | **0.219** | 0.175 | 0.235 | 0.172 | 0.220 | 0.217 | 0.296 | 0.266 | 0.336 | 0.300 | 0.384 | 0.896 | 0.556 | 0.458 | 0.490 |
| | 192 | **0.205** | **0.250** | 0.218 | 0.278 | 0.219 | 0.261 | 0.276 | 0.336 | 0.307 | 0.367 | 0.598 | 0.544 | 0.622 | 0.624 | 0.658 | 0.589 |
| | 336 | **0.251** | **0.300** | 0.263 | 0.314 | 0.280 | 0.306 | 0.339 | 0.380 | 0.359 | 0.395 | 0.578 | 0.523 | 0.739 | 0.753 | 0.797 | 0.652 |
| | 720 | 0.338 | 0.378 | **0.332** | 0.374 | 0.365 | **0.359** | 0.403 | 0.428 | 0.419 | 0.428 | 1.059 | 0.741 | 1.004 | 0.934 | 0.869 | 0.675 |
| | Avg. | **0.238** | **0.286** | 0.247 | 0.300 | 0.259 | 0.287 | 0.309 | 0.360 | 0.338 | 0.382 | 0.634 | 0.548 | 0.815 | 0.717 | 0.696 | 0.601 |

## C    IMPLEMENTATION DETAILS

This section discusses the training details of different Predictors, e.g., NODE, Contiformer, and DLinear, and the Corrector (Neural CDE).

### C.1    NODE

The NODE was used as a Predictor for synthetic datasets. The vector field $\mathbf{f}$ (equation 2) is approximated with a fully connected feedforward neural network. The hyperparameters are given below:

- Batch size: 16
- Learning rate: 0.001
- Vector field $\mathbf{f}$: $FC(100)_2$
- Optimizer: `Adam`

where $FC(100)_2$ denotes a fully connected feedforward neural network with 2 hidden layers each with 100 neurons.

### C.2    CONTIFORMER

The Contiformer uses NODE to build a continuous-time transformer. A full connected feedforward neural network is used as an Encoder to learn the vector field. For integration, we used `odeint` from `torchdiffeq` (Chen, 2018) with Dormand-Prince 5(4) (Dopri5), an adaptive explicit Runge-Kutta method, using relative/absolute tolerances ($\mathtt{rtol} = 10^{-3}$, $\mathtt{atol} = 10^{-6}$). The Contiformer has the following hyperparameters:

- Batch size: 64
- Learning rate: 0.001

- Encoder: $FC(100)_2$

- Optimizer: `Adam`

- Heads (H): 4

- Dimension of Key $(d_k)$, Query $(d_q)$, & Value $(d_v)$ per head = 4

## C.3 DLINEAR

DLinear was proposed first in Zeng et al. (2022) to challenge transformer-based solutions for LTSF datasets. We used the official implementation of DLinear from `Github` [2] without changing any hyperparameters. This repository also includes LTSF datasets, which we used in our experiments.

## C.4 NEURAL CDE

The Neural CDE was used as a Corrector for all Predictors. We used the following hyperparameters for synthetic, physics simulation, and forecasting datasets.

- Batch size: 256 except ILI where 32 was used

- Learning rate: 0.001

- Neural CDE vector field $f_\theta$: $FC(400)_4$

- Neural CDE Decoder $\xi_\varphi$ (Appendix G.5): $FC(400)_4$

- Neural CDE initial hidden state network $\zeta_\phi$: $FC(50)_1$

- Neural CDE hidden state dimension $C$: 11

- Optimizer: `Adam`

The early stopping was used to stop the training of NODE, Contiformer, DLinear, and Neural CDE. The MSE loss function is used to train all Predictors. The solver settings for NODE and Neural CDE are the following. The `diffrax` implementation of `diffeqsolve` is used. For integration, we used `Tsit5` solver (Appendix G.4), a 5th-order explicit Runge-Kutta method with an embedded 4th-order method for adaptive step sizing. The PID controller, with relative/absolute tolerances ($\texttt{rtol} = 10^{-3}$, $\texttt{atol} = 10^{-6}$), is used to control the next step size based on the error estimate. The initial step size is 0.001.

# D  SYNTHETIC DATASETS

We train NODE on four synthetic datasets, shown in Table 1. The closed-form expressions of multivariate ODEs are provided in this section. The parameters and initial conditions are adopted from Shahid & Fleming (2025).

## D.1  LOTKA-VOLTERRA (WANGERSKY, 1978)

$$\frac{dx}{dt} = \alpha x - \beta xy \tag{6}$$

$$\frac{dy}{dt} = \delta xy - \gamma y \tag{7}$$

**Initial Condition Ranges**: $x \in [5, 20]$; $y \in [5, 10]$
**Parameters**: $\alpha = 1.1$; $\beta = 0.4$; $\gamma = 0.4$; $\delta = 0.1$

---

[2]`https://github.com/cure-lab/LTSF-Linear`

### D.2 LORENZ (BRUNTON ET AL., 2016)

$$\frac{dx}{dt} = \sigma(y - x) \tag{8}$$

$$\frac{dy}{dt} = x(\rho - z) - y \tag{9}$$

$$\frac{dz}{dt} = xy - \beta z \tag{10}$$

**Initial Condition Ranges**: $x \in [-20, 20]$; $y \in [-20, 20]$; $z \in [0, 50]$
**Parameters**: $\sigma = 10$; $\rho = 28$; $\beta = \frac{8}{3}$

### D.3 FITZHUGH-NAGUMO (FHN) (IZHIKEVICH & FITZHUGH, 2006A)

$$\frac{dv}{dt} = v - \frac{v^3}{3} - w + I \tag{11}$$

$$\frac{dw}{dt} = \epsilon(v + a - bw) \tag{12}$$

**Initial Condition Ranges**: $v \in [-1.5, 1.5]$; $w \in [-1.5, 1.5]$
**Parameters**: $a = 0.7$; $b = 0.8$; $\epsilon = 0.08$; $I = 0.5$

### D.4 GLYCOLYTIC OSCILLATOR (DANIELS & NEMENMAN, 2015)

$$\frac{dS_1}{dt} = J_0 - \frac{k_1 S_1 S_6}{1 + (S_6/K_1)^q} \tag{13}$$

$$\frac{dS_2}{dt} = 2\frac{k_1 S_1 S_6}{1 + (S_6/K_1)^q} - k_2 S_2(N - S_5) - k_6 S_2 S_5 \tag{14}$$

$$\frac{dS_3}{dt} = k_2 S_2(N - S_5) - k_3 S_3(A - S_6) \tag{15}$$

$$\frac{dS_4}{dt} = k_3 S_3(A - S_6) - k_4 S_4 S_5 - \kappa(S_4 - S_7) \tag{16}$$

$$\frac{dS_5}{dt} = k_2 S_2(N - S_5) - k_4 S_4 S_5 - k_6 S_2 S_5 \tag{17}$$

$$\frac{dS_6}{dt} = -2\frac{k_1 S_1 S_6}{1 + (S_6/K_1)^q} + 2k_3 S_3(A - S_6) - k_5 S_6 \tag{18}$$

$$\frac{dS_7}{dt} = \psi\kappa(S_4 - S_7) - kS_7 \tag{19}$$

**Initial Condition Ranges**: $S_1 \in [0.15, 1.60]$; $S_2 \in [0.19, 2.16]$; $S_3 \in [0.04, 0.20]$; $S_4 \in [0.10, 0.35]$; $S_5 \in [0.08, 0.30]$; $S_6 \in [0.14, 2.67]$; $S_7 \in [0.05, 0.10]$
**Parameters**: $J_0 = 2.5$; $k_1 = 100$; $k_2 = 6$; $k_3 = 16$; $k_4 = 100$; $k_5 = 1.28$; $k_6 = 12$; $k = 1.8$; $\kappa = 13$; $q = 4$; $K_1 = 0.52$; $\psi = 0.1$; $N = 1$; $A = 4$

### D.5 DATA GENERATION DETAILS

Table 10: Details about data generation

| Model | $\Delta t$ | Timesteps | Trajectories |
|---|---|---|---|
| Lotka Volterra | 0.1 | 300 | 500 |
| Lorenz | 0.01 | 300 | 1000 |
| FHN | 0.5 | 400 | 350 |
| Glycolytic | 0.01 | 400 | 750 |

## E    PHYSICS SIMULATION DATASETS

To collect the MuJoCo dataset, we trained expert policies for each environment and used them to generate trajectory rollouts. The policies were deterministic because the focus of this study is on learning the evolution of states in the system. Deterministic controllers ensure consistent state transitions across rollouts, whereas stochastic controllers could produce different trajectories from the same initial condition, which would complicate modeling when only the states are used as inputs. An extension to action-conditioned dynamics models is natural and would involve learning a mapping of the form $\mathbf{x}(t+1) = g_\theta(\mathbf{x}(t), \mathbf{a}(t))$ instead of $\mathbf{x}(t+1) = g_\theta(\mathbf{x}(t))$, where $\mathbf{x}(t+1)$ denotes the next state transitioned from $\mathbf{x}(t)$ by applying action $\mathbf{a}(t)$. Such a formulation would allow the use of data from arbitrary policies, including stochastic ones, and will be considered in future work. For each environment, we generated 2000 trajectories of 300 regularly-sampled time points each. There are four MuJoCo environments in Table.2. Fig. 5 shows the four environments within the simulator.

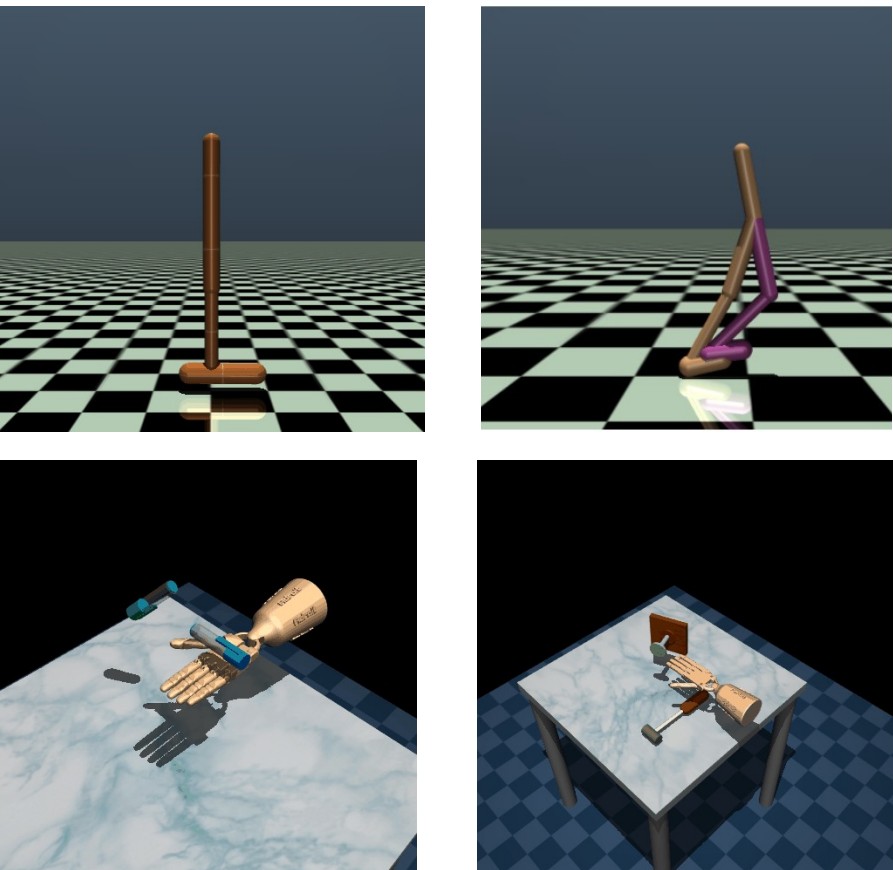

Figure 5: The MuJoCo environments inside the simulator **(a)** Upper left: Hopper (11D) **(b)** Upper right: Walker2D (17D) **(c)** Lower left: Pen (45D) **(d)** Lower right: Hammer (46D)

## F    PERFORMANCE OF PREDICTOR-CORRECTOR ON PEN (45D)

The performance of the ContiFormer without Corrector (w/o) and with Corrector (w/) on Pen is shown in Table 2. Here, we show the performance of Corrector on one of the trajectories from the test dataset for a 20% observed points setting. The first 24 dimensions are shown in Fig. 6 and the rest of the 21 dimensions in Fig. 7. The visualizations demonstrate that the Corrector (trained on the first 50 timesteps) can correct the Predictor up to 200 timesteps, well beyond the training horizon, for such a high-dimensional dynamical system.

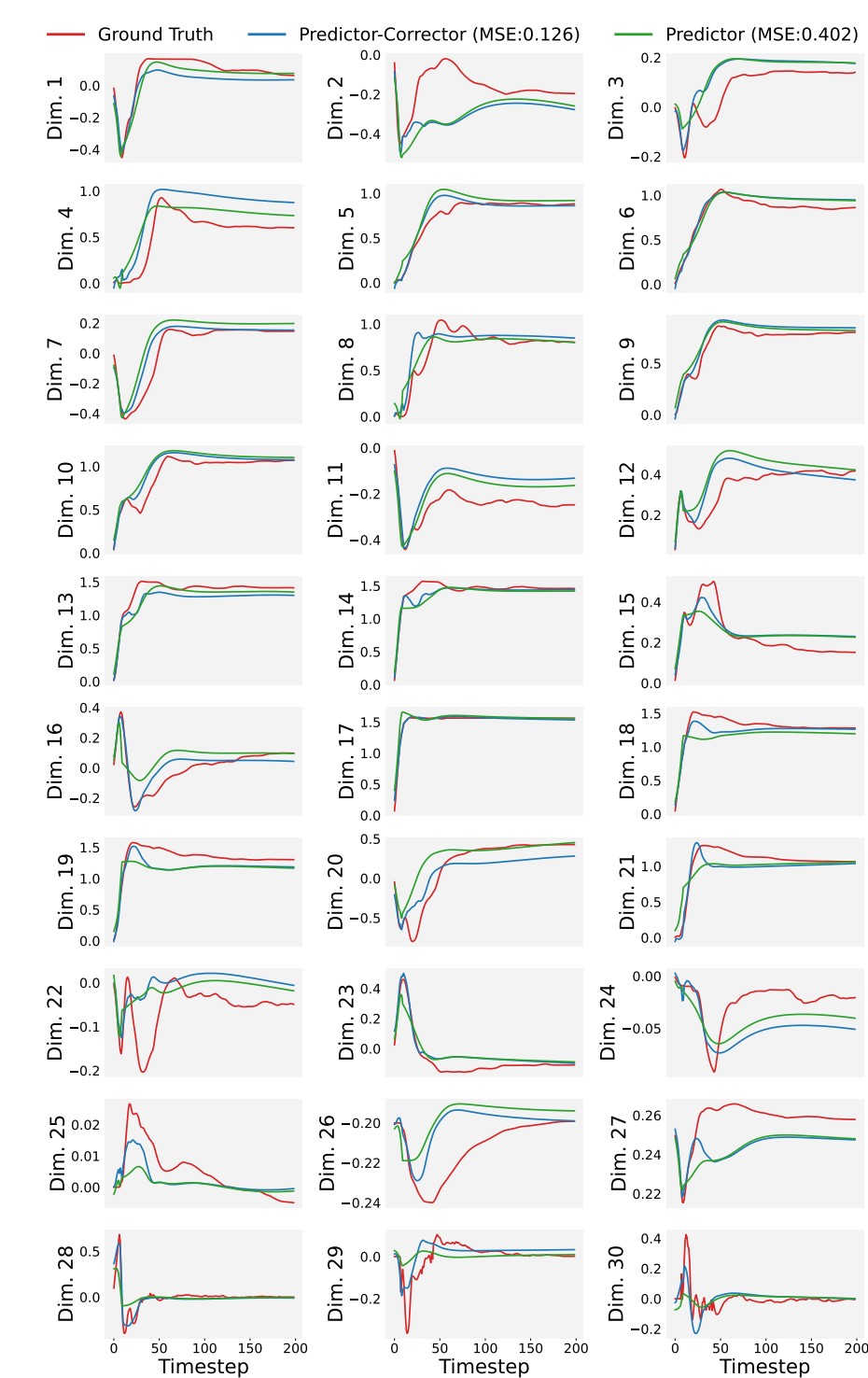

Figure 6: The performance of Corrector on one of the trajectories of Pen for a 20% observed points setting. The first 30 dimensions of the Pen trajectory are shown here. Dim. stands for dimension.

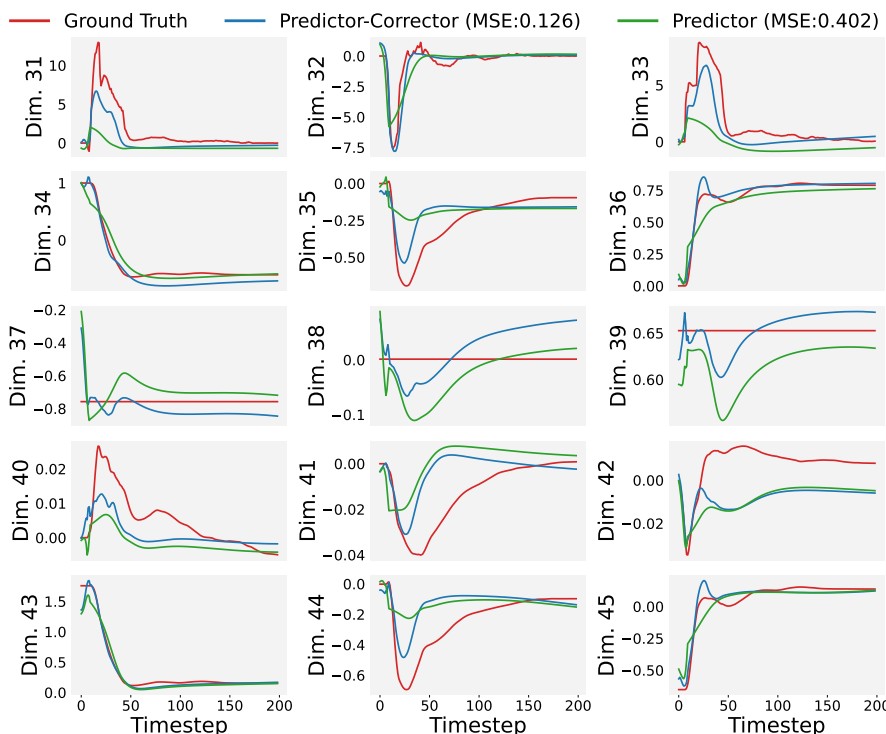

Figure 7: The performance of Corrector on one of the trajectories of Pen for a 20% observed points setting. The last 15 dimensions of the Pen trajectory are shown here. Dim. stands for dimension.

# G   ABLATION STUDIES

## G.1   TRAINING HORIZON FOR LTSF

In section 5.5, we discussed that the Corrector is trained on the first 50, 100, 150, and 300 timesteps for forecast horizons 96, 192, 336, and 720, respectively. Here, we chose forecast horizon 336 for Exchange, ETTm2, ETTh2, & Weather to demonstrate the impact of training Corrector on the first 50, 100, 150, 200, 250, 300, and 336 timesteps. Table 11 shows that the Corrector shows a relatively poor performance, i.e., an increase in MSE/MAE, when trained on very short (e.g., 50 timesteps) or very long horizons (e.g., 336 timesteps). This motivates our choice of training the Corrector on intermediate-length horizons (e.g., 50 for $T$=96, 100 for $T$=192, etc.) to report results in Table 4 and 9.

Table 11: Multivariate LTSF errors (MSE/MAE) for forecast horizon 336 of Exchange, ETTm2, ETTh2, & Weather, where the Corrector is trained on the first 50, 100, 150, 200, 250, 300, & 336 timesteps for each setting. The %↓ shows the reduction in MSE in percentage.

| Train Horizon | 50 | | 100 | | 150 | | 200 | | 250 | | 300 | | 336 | |
|---|---|---|---|---|---|---|---|---|---|---|---|---|---|---|
| Metrics | MSE | MAE | MSE | MAE | MSE | MAE | MSE | MAE | MSE | MAE | MSE | MAE | MSE | MAE |
| Exchange | 0.304 | 0.413 | 0.278 | 0.394 | 0.243 | 0.370 | 0.305 | 0.434 | 0.353 | 0.470 | 0.340 | 0.460 | 0.360 | 0.472 |
| | 8.71%↓ | 6.56%↓ | 16.51%↓ | 10.86%↓ | 27.03%↓ | 16.29%↓ | 8.41%↓ | 1.81%↓ | -6.01%↓ | -6.3%↓ | -2.10%↓ | -4.07%↓ | -8.11%↓ | -6.79%↓ |
| ETTh2 | 0.463 | 0.473 | 0.459 | 0.470 | 0.458 | 0.470 | 0.456 | 0.469 | 0.459 | 0.470 | 0.460 | 0.471 | 0.471 | 0.480 |
| | 0.64%↓ | 0.42%↓ | 1.50%↓ | 1.05%↓ | 1.72%↓ | 1.05%↓ | 2.15%↓ | 1.26%↓ | 1.50%↓ | 1.05%↓ | 1.29%↓ | 0.84%↓ | -1.07%↓ | -1.05%↓ |
| ETTm2 | 0.288 | 0.353 | 0.287 | 0.352 | 0.288 | 0.354 | 0.290 | 0.356 | 0.293 | 0.358 | 0.291 | 0.355 | 0.292 | 0.357 |
| | 2.37%↓ | 1.67%↓ | 2.71%↓ | 1.95%↓ | 2.37%↓ | 1.39%↓ | 1.69%↓ | 0.84%↓ | 0.68%↓ | 0.28%↓ | 0.68%↓ | 0.28%↓ | 1.02%↓ | 0.56%↓ |
| Weather | 0.264 | 0.320 | 0.253 | 0.306 | 0.251 | 0.300 | 0.253 | 0.305 | 0.254 | 0.303 | 0.256 | 0.306 | 0.251 | 0.300 |
| | -0.38%↓ | -1.91%↓ | 3.80%↓ | 2.55%↓ | 4.56%↓ | 3.82%↓ | 3.80%↓ | 2.87%↓ | 3.42%↓ | 4.08%↓ | 3.42%↓ | 2.55%↓ | 4.56%↓ | 4.46%↓ |

## G.2 ORDER OF ODE

To show the robustness of Predictor-Corrector to different orders of ODEs. We generated synthetic data from second, third, and fourth-order ODEs.

**Second-order system.**    We simulate the damped oscillator

$$x''(t) + 0.3\,x'(t) + 1.0\,x(t) = 0. \tag{20}$$

Its first-order state-space representation is

$$\begin{aligned}
z_1 &= x, \\
z_2 &= x', \\
\dot{z}_1 &= z_2, \\
\dot{z}_2 &= -1.0\,z_1 - 0.3\,z_2.
\end{aligned} \tag{21}$$

**Third-order system.**    We simulate the third-order linear ODE

$$x^{(3)}(t) + 0.4\,x''(t) + 0.3\,x'(t) + 1.0\,x(t) = 0. \tag{22}$$

Its first-order state-space form is

$$\begin{aligned}
z_1 &= x, \\
z_2 &= x', \\
z_3 &= x'', \\
\dot{z}_1 &= z_2, \\
\dot{z}_2 &= z_3, \\
\dot{z}_3 &= -1.0\,z_1 - 0.3\,z_2 - 0.4\,z_3.
\end{aligned} \tag{23}$$

**Fourth-order system.**    We simulate the fourth-order ODE

$$x^{(4)}(t) + 0.3\,x^{(3)}(t) + 0.5\,x''(t) + 0.3\,x'(t) + 1.0\,x(t) = 0. \tag{24}$$

The corresponding first-order state-space form is

$$\begin{aligned}
z_1 &= x, \\
z_2 &= x', \\
z_3 &= x'', \\
z_4 &= x^{(3)}, \\
\dot{z}_1 &= z_2, \\
\dot{z}_2 &= z_3, \\
\dot{z}_3 &= z_4, \\
\dot{z}_4 &= -1.0\,z_1 - 0.3\,z_2 - 0.5\,z_3 - 0.3\,z_4.
\end{aligned} \tag{25}$$

We train NODE as a Predictor for these examples with the same architectural details given in C.1. Both states and derivatives are modeled with NODE. The Corrector is trained with the same architectural details given in C.4. To simulate time-varying partial observability, we randomly mask half of the observed features for 0%, 50%, and 80% of the observed time points. The results are shown in the Table. 12. The interpolation shows the performance of the Corrector for the first 50 timesteps, which corresponds to its training horizon. The extrapolation columns lists the timesteps for every

Table 12: Test MSE of NODE as a Predictor on ODEs of different orders with (w/) and without (w/o) Corrector. The % ↓ shows the percentage reduction in MSE of NODE with our proposed Corrector. For both interpolation and extrapolation, reported MSE values are computed from timestep 0 up to the specified timestep ($t$) for each setting (0-$t$).

| Dynamical System | Model | Interpolation (% Pts w/ missing features) | | | Extrapolation (% Pts w/ missing features) | | |
|---|---|---|---|---|---|---|---|
| | | 0% | 30% | 60% | 0% | 30% | 60% |
| **2nd Order ODE** | w/o | 0.0142 | 0.0452 | 0.0728 | 0.0413 | 0.0952 | 0.109 |
| | w/ | 0.0039 | 0.0152 | 0.0252 | 0.0397 | 0.0922 | 0.097 |
| | $0-t$ \| %↓ | 0–50 \| 72% | 0–50 \| 66% | 0–50 \| 65% | 0–170 \| 4% | 0–180 \| 3% | 0–165 \| 10% |
| **3rd Order ODE** | w/o | 0.0101 | 0.0359 | 0.0934 | 7.3 | 8.1 | 9.4 |
| | w/ | 0.0019 | 0.0039 | 0.0543 | 6.9 | 7.9 | 9.0 |
| | $0-t$ \| %↓ | 0–50 \| 81% | 0–50 \| 89% | 0–50 \| 41% | 0–180 \| 5% | 0–170 \| 4% | 0–190 \| 4% |
| **4th Order ODE** | w/o | 1.27 | 2.3 | 4.1 | 10.36 | 12.45 | 14.56 |
| | w/ | 1.15 | 2.1 | 3.8 | 10.02 | 11.9 | 14.10 |
| | $0-t$ \| %↓ | 0–50 \| 9% | 0–50 \| 8% | 0–50 \| 7% | 0–90 \| 3% | 0–80 \| 4% | 0–100 \| 3% |

Table 13: MSE of ContiFormer as a Predictor on various dynamical systems' datasets from MuJoCo (Todorov et al., 2012) with (w/) and without (w/o) Corrector under different interpolation schemes for control paths of Neural CDE. For interpolation and extrapolation, the reported MSE values are computed from timestep 0 up to the specified timestep ($t$) for each setting (0-$t$).

| Dynamical System (Interpolation) | Model | Interpolation (% Observed Pts) | | | | Extrapolation (% Observed Pts) | | | |
|---|---|---|---|---|---|---|---|---|---|
| | | 20% | 50% | 80% | 100% | 20% | 50% | 80% | 100% |
| **Walker2D** (Linear) | w/o | 1.826 | 0.572 | 0.408 | 0.164 | 2.72 | 0.577 | 0.778 | 0.869 |
| | w/ | 1.029 | 0.435 | 0.228 | 0.073 | 2.62 | 0.532 | 0.758 | 0.839 |
| | $0-t$ \| %↓ | 0–50 \| 43% | 0–50 \| 23% | 0–50 \| 43% | 0–50 \| 55% | 0–180 \| 4% | 0–100 \| 7% | 0–130 \| 3% | 0–140 \| 3% |
| **Walker2D** (Cubic) | w/o | 1.826 | 0.572 | 0.408 | 0.164 | 2.89 | 1.70 | 0.778 | 0.869 |
| | w/ | 0.721 | 0.285 | 0.149 | 0.053 | 2.68 | 1.63 | 0.750 | 0.838 |
| | $0-t$ \| %↓ | 0–50 \| 61% | 0–50 \| 50% | 0–50 \| 63% | 0–50 \| 68% | 0–190 \| 7% | 0–180 \| 4% | 0–130 \| 4% | 0–140 \| 4% |
| **Hammer** (Linear) | w/o | 0.020 | 0.0137 | 0.0106 | 0.0085 | 0.0115 | 0.0083 | 0.0049 | 0.0046 |
| | w/ | 0.0122 | 0.0058 | 0.0038 | 0.0039 | 0.0110 | 0.0080 | 0.0047 | 0.0045 |
| | $0-t$ \| %↓ | 0–50 \| 39% | 0–50 \| 57% | 0–50 \| 63% | 0–50 \| 54% | 0–140 \| 4% | 0–140 \| 3% | 0–180 \| 4% | 0–150 \| 6% |
| **Hammer** (Cubic) | w/o | 0.020 | 0.0137 | 0.0106 | 0.0085 | 0.0158 | 0.0073 | 0.0049 | 0.0046 |
| | w/ | 0.012 | 0.0059 | 0.0037 | 0.0032 | 0.0150 | 0.0070 | 0.0047 | 0.0044 |
| | $0-t$ \| %↓ | 0–50 \| 37% | 0–50 \| 56% | 0–50 \| 64% | 0–50 \| 63% | 0–100 \| 4% | 0–150 \| 4% | 0–180 \| 4% | 0–150 \| 6% |

setting up to which the Corrector brings at least a 3% reduction in MSE of NODE. The corrector consistently brings a reduction in the MSE of the Predictor irrespective of the levels of partial observability. However, the order of the ODE does impact the performance of the Corrector. The Corrector brings less than 10% reduction in MSE within the interpolation region for the 4th order system compared to the 2nd and 3rd order systems, where the reduction in MSEs is significantly higher. Within the extrapolation region, the horizons up to which the Corrector brings at least 3% reduction in MSE are much smaller for 4th order systems compared to 2nd and 3rd order systems. This demonstrates that the order of the system impacts the performance of the Corrector.

## G.3 SENSITIVITY TO INTERPOLATION SCHEMES

The `diffrax` Kidger (2022) contains different interpolation schemes for control paths of Neural CDE. Here, we present the sensitivity of Corrector performance to those interpolation schemes. The results with Cubic interpolation for Walker2D and Hammer are copied from the Table 2. The Linear interpolation brings a slightly smaller reduction in MSE (%) compared to the Cubic interpolation in almost every setting in Table 13. This observation is in line with the results reported by Morrill et al. (2022).

## G.4 SENSITIVITY TO ODE SOLVERS

The `diffrax` package has different explicit Runga-Kutta (RK) methods. We used `Tsit5` to report results in the paper everywhere else. Here, we analyze the sensitivity of Walker2D results to other solvers from the explicit RK family. The results are reported in Table 14. The `Tsit5` performed better both in interpolation and extrapolation regions compared to other solvers, followed by `Dopri5`. The adaptive-step size solvers, such as `Heun`, `Dopri5`, & `Tsit5`, performed better than `Euler` method.

Table 14: MSE of ContiFormer as a Predictor on Walker2D dataset from MuJoCo (Todorov et al., 2012) with (w/) and without (w/o) Corrector under different solvers for Neural CDE. For interpolation and extrapolation, the reported MSE values are computed from timestep 0 up to the specified timestep ($t$) for each setting (0-$t$).

| ODE Solver | Model | Interpolation (% Observed Pts) | | | | Extrapolation (% Observed Pts) | | | |
|---|---|---|---|---|---|---|---|---|---|
| | | 20% | 50% | 80% | 100% | 20% | 50% | 80% | 100% |
| Euler | w/o | 1.826 | 0.572 | 0.408 | 0.164 | 2.72 | 0.639 | 0.778 | 0.245 |
| | w/ | 1.375 | 0.437 | 0.292 | 0.105 | 2.62 | 0.620 | 0.743 | 0.234 |
| | 0–$t$ \| %↓ | 0–50 \| 24% | 0–50 \| 23% | 0–50 \| 28% | 0–50 \| 36% | 0–180 \| 4% | 0–100 \| 3% | 0–130 \| 4% | 0–80 \| 4% |
| Heun | w/o | 1.826 | 0.572 | 0.408 | 0.164 | 1.86 | 0.969 | 0.778 | 0.413 |
| | w/ | 0.901 | 0.359 | 0.328 | 0.078 | 1.67 | 0.925 | 0.752 | 0.394 |
| | 0–$t$ \| %↓ | 0–50 \| 50% | 0–50 \| 37% | 0–50 \| 19% | 0–50 \| 52% | 0–130 \| 9% | 0–130 \| 4% | 0–130 \| 4% | 0–120 \| 4% |
| Dopri5 | w/o | 1.826 | 0.572 | 0.408 | 0.164 | 1.86 | 0.969 | 0.778 | 0.869 |
| | w/ | 0.819 | 0.361 | 0.329 | 0.064 | 1.81 | 0.929 | 0.760 | 0.841 |
| | 0–$t$ \| %↓ | 0–50 \| 55% | 0–50 \| 36% | 0–50 \| 19% | 0–50 \| 60% | 0–130 \| 3% | 0–130 \| 4% | 0–130 \| 3% | 0–140 \| 4% |
| Tsit5 | w/o | 1.826 | 0.572 | 0.408 | 0.164 | 2.89 | 1.70 | 0.778 | 0.869 |
| | w/ | 0.721 | 0.285 | 0.149 | 0.053 | 2.68 | 1.63 | 0.750 | 0.838 |
| | 0–$t$ \| %↓ | 0–50 \| 61% | 0–50 \| 50% | 0–50 \| 63% | 0–50 \| 68% | 0–190 \| 7% | 0–180 \| 4% | 0–130 \| 4% | 0–140 \| 4% |

## G.5 DECODER ($\xi_\varphi$) SIZE

The decoder ($\xi_\varphi$) for Neural CDE corrector maps the hidden state dynamics $\mathbf{z}(t)$ to error dynamics $\hat{\mathbf{e}}(t)$ as shown in Fig. 2. Here, we show the results of FHN with varying decoder sizes. The results elsewhere are reported using a four-layer fully connected neural network with 400 neurons in each layer, denoted as $FC(400)_4$. With $FC(20)_1$, we observe a performance degradation within the interpolation region, but the extrapolation performance is better. This indicates that a small decoder mitigates overfitting and facilitates generalization, while the hidden state $\mathbf{z}(t)$ models error dynamics.

Table 15: MSE of NODE as a Predictor on FHN dataset with (w/) and without (w/o) Corrector under different decoder sizes for Neural CDE (Corrector). For interpolation and extrapolation, the reported MSE values are computed from timestep 0 up to the specified timestep ($t$) for each setting (0-$t$).

| Decoder Size | Model | Interpolation (% Observed Pts) | | | | Extrapolation (% Observed Pts) | | | |
|---|---|---|---|---|---|---|---|---|---|
| | | 20% | 50% | 80% | 100% | 20% | 50% | 80% | 100% |
| $FC(20)_1$ | w/o | 0.225 | 0.161 | 0.150 | 0.137 | 0.264 | 0.183 | 0.200 | 0.180 |
| | w/ | 0.187 | 0.139 | 0.122 | 0.114 | 0.248 | 0.177 | 0.191 | 0.171 |
| | 0 – $t$ \| %↓ | 0–50 \| 17% | 0–50 \| 14% | 0–50 \| 18% | 0–50 \| 17% | 0–90 \| 5% | 0–190 \| 4% | 0–220 \| 5% | 0–200 \| 5% |
| $FC(100)_1$ | w/o | 0.225 | 0.161 | 0.150 | 0.137 | 0.274 | 0.178 | 0.192 | 0.179 |
| | w/ | 0.181 | 0.137 | 0.123 | 0.107 | 0.263 | 0.169 | 0.168 | 0.173 |
| | 0 – $t$ \| %↓ | 0–50 \| 18% | 0–50 \| 12% | 0–50 \| 18% | 0–50 \| 22% | 0–100 \| 4% | 0–150 \| 4% | 0–200 \| 4% | 0–180 \| 4% |
| $FC(400)_4$ | w/o | 0.225 | 0.161 | 0.150 | 0.137 | 0.242 | 0.178 | 0.192 | 0.166 |
| | w/ | 0.164 | 0.128 | 0.093 | 0.097 | 0.231 | 0.171 | 0.183 | 0.158 |
| | 0 – $t$ \| %↓ | 0–50 \| 27% | 0–50 \| 20% | 0–50 \| 38% | 0–50 \| 29% | 0–75 \| 5% | 0–150 \| 4% | 0–140 \| 4% | 0–150 \| 5% |

### G.6 EFFICIENCY–EXTRAPOLATION PARETO CURVES FOR $\kappa$ AND $\eta$ REGULARIZATION

The $\kappa$ (Sparse Control Paths) and $\eta$ (Variable Length Control Paths) regularization strategies are two proposed approaches to enhance the extrapolation performance and computational efficiency (i.e., NFE) of the Neural CDE corrector. Both approaches offer a trade-off between the efficiency, i.e., number of function evaluations (NFEs), and the extrapolation horizon. The extrapolation horizon is the timestep up to which the corrector brings at least 3% reduction in MSE. We plot Pareto curves for both $\kappa$ and $\eta$ to show the trade-off between NFE and extrapolation horizon in Fig. 8. Pareto curves for $\kappa$ & $\eta$ show the value of $\kappa$ & $\eta$ right next to each data point, while the x and y axes show the NFE and extrapolation horizon, respectively. It can be observed that achieving both a small NFE and a large extrapolation horizon is a challenging task. The best point is the one that balances both NFE and the extrapolation horizon.

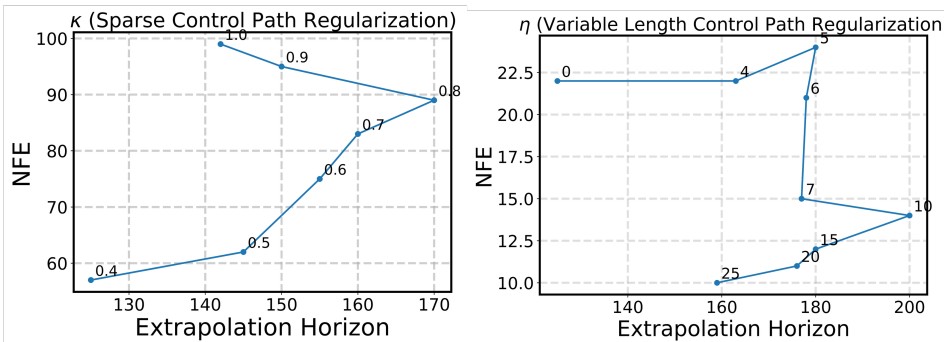

Figure 8: Pareto curves showing the trade-offs between extrapolation and efficiency via the proposed regularization strategies.

### G.7 TWO-STAGE VERSUS ALTERNATE TRAINING

The Predictors and Correctors are trained in two different stages elsewhere in the paper. The two-stage training involves the Predictor first until convergence and the Corrector learns the error dynamics of the Predictor afterwards. Another way is to train them jointly. A natural way of joint-training is to train them both alternatively where Predictor and Corrector do gradient updates alternatively. The Predictor takes a few gradient steps and the Corrector does a few gradient updates to learn the updated error dynamics of Predictor. This essentially creates a moving target for the Corrector and made the training less stable compared to two-stage training. The alternate paradigm lowers the bias marginally and results in higher variance, which manifests in the form of oscillatory training. Empirically, we did not observe any improvement in extrapolation with alternate training and believe that two-stage training achieves a better bias-variance-stability trade-off.

Table 16: Test MSE of NODE as a Predictor on Lorenz dataset with (w/) and without (w/o) Corrector. The results from Table 1 (two-stage training) and alternate training are compared. We don't simulate any irregular sampling during training. The % ↓ shows the percentage reduction in MSE of NODE with our proposed Corrector. For both interpolation and extrapolation, reported MSE values are computed from timestep 0 up to the specified timestep ($t$) for each setting (0-$t$).

| Dynamical System | Model | Interpolation | | Extrapolation | |
|---|---|---|---|---|---|
| | | Two-stage | Alternate | Two-stage | Alternate |
| Lorenz | w/o | 0.887 | 0.870 | 6.464 | 4.102 |
| | w/ | 0.468 | 0.625 | 6.245 | 3.901 |
| | $0-t$ \| %↓ | 0–50 \| 47% | 0–50 \| 28% | 0–150 \| 3% | 0–120 \| 4% |

### G.8 WALL-CLOCK TIME OF $\kappa$ & $\eta$

We report the average wall-clock time it takes to complete one epoch during training, varying the levels of $\kappa$ & $\eta$, as shown in Fig. 9. It can be seen that smaller values of $\kappa$ and larger values of $\eta$ result in faster training, corroborating the results shown in Fig. 4 of the paper.

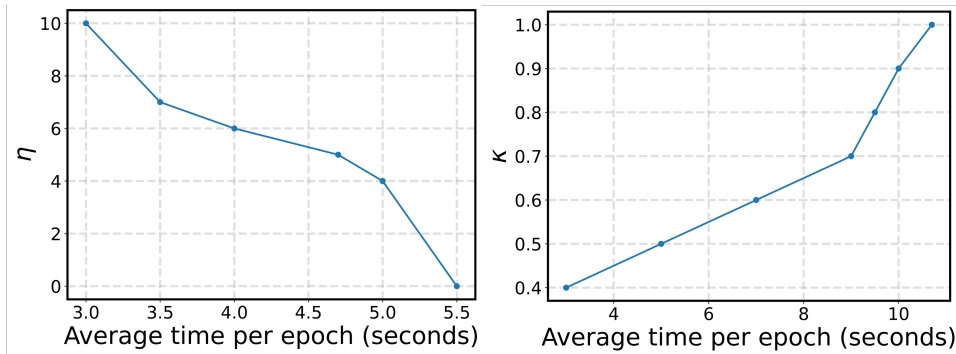

Figure 9: Average wall-clock time (in seconds) of an epoch with varying values of $\kappa$ & $\eta$.

### G.9 LONG-HORIZON STRESS TESTS

We empirically show that the MSE of corrected forecasts of Predictor over long horizons (400 timesteps) remains well-bounded. The results are shown for Lorenz, FHN, LVolt, and Glycolytic in the Fig. 10. Each data point shows the log(MSE) from timestep 0 to timestep T on the x-axis.

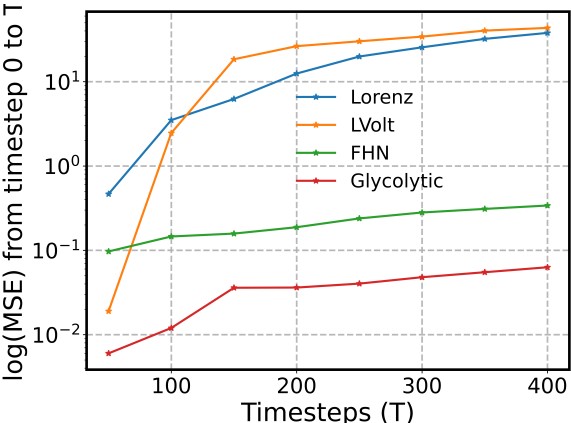

Figure 10: The long-horizon tests demonstrating the well-boundedness of the error of the corrected forecasts of the Predictor (NODE) on Lorenz, LVolt, FHN, & Glycolytic.

## H ADDITIONAL RESULTS

### H.1 ADDITIONAL LTSF RESULTS

We add two baselines (i.e., MLP and Diffusion (Sohl-Dickstein et al., 2015)) as correctors to compare against the Neural CDE corrector on the Exchange LTSF dataset. The LTSF datasets are regularly sampled; therefore, it is reasonable to utilize MLP and Diffusion as correctors. The results are given in Table 17. This ablation study was conducted to demonstrate the competitiveness of the Neural CDE corrector against MLP and Diffusion on regularly sampled LTSF datasets. This establishes Neural CDE as a unified Corrector for continuous- and discrete-time Predictors and regularly/irregularly sampled time series.

Table 17: Multivariate long-term forecasting errors (MSE/MAE; lower is better). Best value in each row is highlighted in bold.

| Methods | | DLinear (w/) Neural CDE | | DLinear (w/) MLP | | DLinear (w/) Diffusion | | DLinear (w/o) | |
|---|---|---|---|---|---|---|---|---|---|
| Metric | | MSE | MAE | MSE | MAE | MSE | MAE | MSE | MAE |
| Exchange | 96 | 0.083 | 0.207 | 0.084 | 0.205 | **0.071** | **0.195** | 0.085 | 0.210 |
| | 192 | 0.159 | 0.295 | 0.159 | 0.296 | **0.152** | **0.284** | 0.162 | 0.297 |
| | 336 | **0.243** | **0.370** | 0.301 | 0.399 | 0.301 | 0.398 | 0.333 | 0.442 |
| | 720 | **0.482** | **0.548** | 0.692 | 0.698 | 0.576 | 0.599 | 0.896 | 0.724 |
| | Avg. | **0.242** | **0.355** | 0.309 | 0.400 | 0.275 | 0.369 | 0.369 | 0.418 |

To demonstrate the effectiveness of Neural CDE on a transformer-based model for LTSF, we test Neural CDE to improve the performance of FEDformer on the ETTm2 dataset. The results are shown in the Table. 18.

Table 18: Multivariate long-term forecasting errors (MSE/MAE; lower is better). The results of FEDformer with (w/) and without (w/o) our Corrector on ETTm2 are shown.

| Methods | | FEDformer (w/) | | FEDformer (w/o) | |
|---|---|---|---|---|---|
| Metric | | MSE | MAE | MSE | MAE |
| ETTm2 | 96 | 0.189 | 0.265 | 0.203 | 0.287 |
| | 192 | 0.250 | 0.311 | 0.269 | 0.328 |
| | 336 | 0.315 | 0.356 | 0.325 | 0.366 |
| | 720 | 0.415 | 0.405 | 0.421 | 0.415 |
| | Avg. | 0.292 | 0.334 | 0.304 | 0.349 |

## H.2 PREDICTOR DIVERSITY

To bolster our claim that the proposed Predictor-Corrector framework is agnostic to the underlying Predictor, we evaluate our Corrector on two additional Predictors, i.e., RNN and TCN. The datasets are Lorenz and FHN. The extrapolation columns list the horizon up to which the Corrector brings at least 3% reduction in MSE of Predictor.

Table 19: Test MSE of RNN & TCN as Predictors on Lorenz & FHN dataset with (w/) and without (w/o) Corrector. The % ↓ shows the percentage reduction in MSE of RNN & TCN with our proposed Corrector. For both interpolation and extrapolation, reported MSE values are computed from timestep 0 up to the specified timestep ($t$) for each setting (0-$t$).

| Dynamical System | Model | Interpolation | | Extrapolation | |
|---|---|---|---|---|---|
| | | RNN | TCN | RNN | TCN |
| Lorenz | w/o | 1.102 | 0.725 | 5.705 | 3.908 |
| | w/ | 0.901 | 0.635 | 5.504 | 3.709 |
| | $0-t$ \| %↓ | 0–50 \| 18% | 0–50 \| 13% | 0–80 \| 3% | 0–100 \| 5% |
| Glycolytic Oscillator | w/o | 0.190 | 0.140 | 0.187 | 0.160 |
| | w/ | 0.140 | 0.110 | 0.180 | 0.150 |
| | $0-t$ \| %↓ | 0–50 \| 26% | 0–50 \| 21% | 0–90 \| 4% | 0–110 \| 6% |

## I  PERFORMANCE OF PREDICTOR-CORRECTOR ON EXCHANGE (8D)

Table 4 shows the performance of DLinear without Corrector (w/o) and with Corrector (w/). Fig. 11 shows the performance of Predictor-Corrector compared to Predictor alone on one of the trajectories from the Exchange test dataset. The trajectory shows a drop in MSE with Corrector.

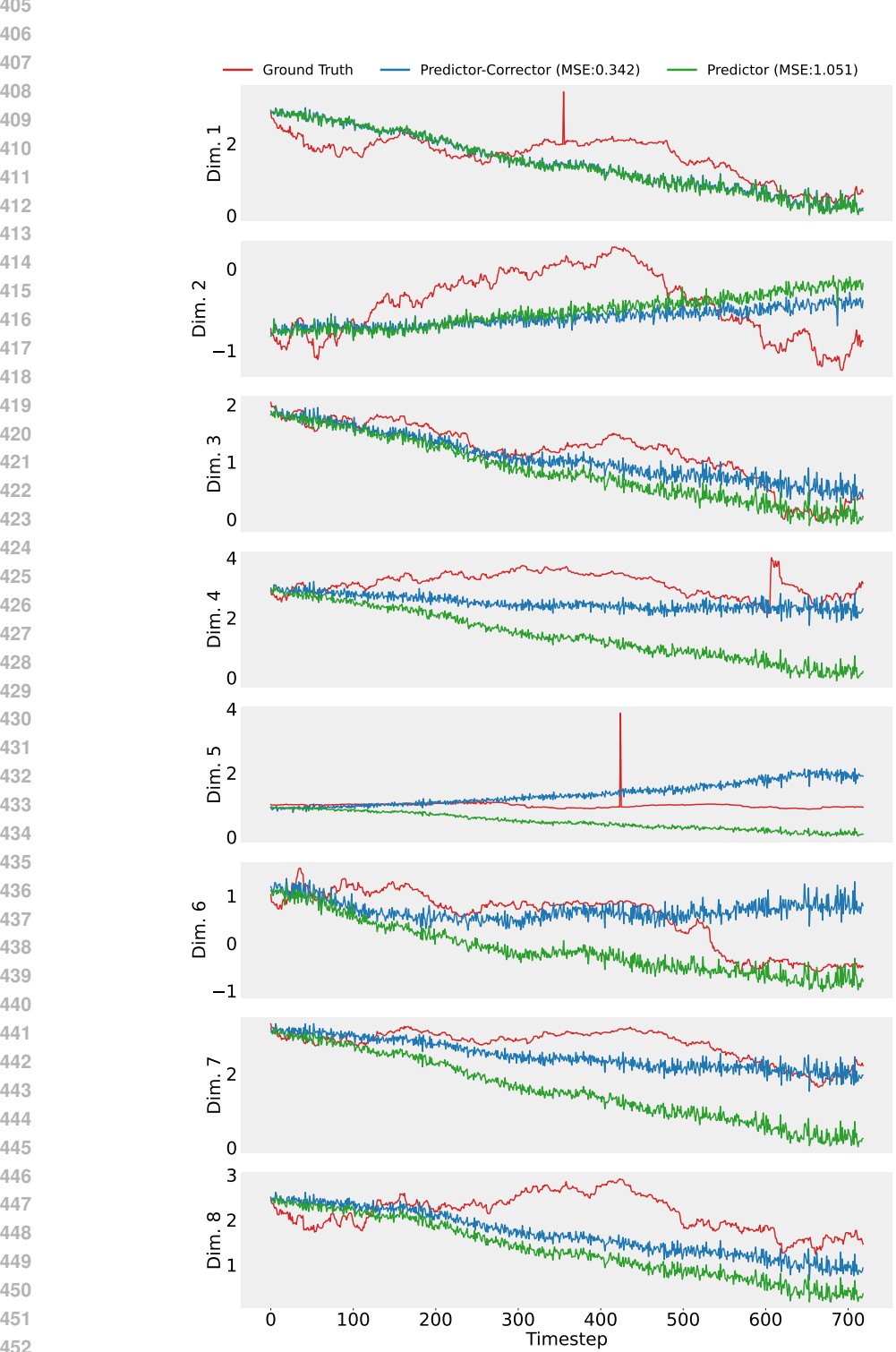

Figure 11: The performance of Corrector on one of the trajectories of the Exchange test dataset. Dim. is used as a shorthand for dimension in the plots.