# OpenReview forum: "Neural CDEs as Correctors for learned time series models"
_ICLR.cc/2026/Conference — Submitted to ICLR 2026_

### Official Review · Reviewer_kUzq · 2025-10-30

**Soundness:** 3
**Presentation:** 4
**Contribution:** 2
**Rating:** 4
**Confidence:** 3

**Summary:**

This paper introduces a general predictor–corrector framework for enhancing the performance of learned time-series models. The corrector is implemented via a Neural Controlled Differential Equation that models the dynamics of prediction errors driven by the predictor’s forecast trajectories, augmented with two regularization strategies to improve extrapolation. Experiments across diverse datasets demonstrate that the proposed method consistently improves the performance of baseline predictors.

**Strengths:**

1.  The writing is clear.
2.  The proposed method is easy to understand and implement. It can be integrated with any time series models conveniently.

**Weaknesses:**

1. There are potential fairness concerns in the experimental comparisons, particularly in the LTSF section. The study primarily contrasts DLinear augmented with the proposed Corrector against various Transformer baselines, but does not evaluate the Corrector when attached to strong Transformer models. Transformer-based architectures such as iTransformer [1] and PatchTST [2] already outperform DLinear; consequently, it remains unclear whether the proposed method can still deliver gains when applied to competitive state-of-the-art models.
2. Since the additional NCDE is introduced as the corrector, it is important to compare against alternative baselines to substantiate the advantage of NCDE. For example, what performance can be achieved by augmenting ContiFormer with an additional linear head to model the residuals? Does NCDE consistently outperform such a simple baseline? Moreover, within the continuous-time paradigm, the paper does not compare against other plausible continuous-time correctors (e.g., using ODE-RNN [3], GRU-ODE [4], or Latent SDE [5] to model the error dynamics).
3. Insufficient theoretical analysis: While the motivation is intuitive, the paper lacks a formal treatment of the key assumption that “the error dynamics are sufficiently controlled by the forecast trajectories.” Can relevant literature be cited to support this premise? Why are forecast trajectories used as the sole control signal rather than augmenting the control with “forecasts + observations” or “forecasts + uncertainty estimates” (e.g., predictive variance)? Moreover, there is no analysis of stability, error propagation bounds, or conditions for extrapolative generalization. Could the authors provide sufficient conditions (e.g., under Lipschitz continuity or local linearization) ensuring convergence or boundedness of the error CDE, or derive upper bounds on the corrected forecast error?
4. If the predicted residual has the opposite sign to the true residual, the error may be further amplified. Are there regimes in which the Corrector systematically amplifies errors (e.g., specific phase regions of chaotic systems)? How pronounced is the impact of such cases on overall performance?

---
[1] Liu, Yong, et al. "iTransformer: Inverted Transformers Are Effective for Time Series Forecasting." The Twelfth International Conference on Learning Representations.

[2] Nie, Yuqi, et al. "A Time Series is Worth 64 Words: Long-term Forecasting with Transformers." The Eleventh International Conference on Learning Representations.

[3] Rubanova, Yulia, Ricky TQ Chen, and David K. Duvenaud. "Latent ordinary differential equations for irregularly-sampled time series." Advances in neural information processing systems 32 (2019).

[4] De Brouwer, Edward, et al. "GRU-ODE-Bayes: Continuous modeling of sporadically-observed time series." Advances in neural information processing systems 32 (2019).

[5] Li, Xuechen, et al. "Scalable gradients for stochastic differential equations." International Conference on Artificial Intelligence and Statistics. PMLR, 2020.

**Questions:**

Pls refer to Weaknesses

---

> ### Author Response · Authors · 2025-11-20
> **Comment [1/2]**
>
> **1.** We thank the reviewer for this question. We anticipated this question; hence, we included diverse Predictors in the study. Particularly, we included the Contiformer (continuous-time transformer) to demonstrate that our Corrector can model the error dynamics of a transformer model. While we admit that we tested Contiformer on MuJoCo datasets, and the reviewer is concerned about LTSF datasets, it is infeasible to test every model with different types of datasets. We selected different datasets and Predictors to show the efficacy of our Corrector across different problems and Predictors. That being said, we improve the FEDformer (a transformer variant) for LTSF with our Corrector on ETTm2. The results are shown below.
>
> | Dataset | Horizon | FEDformer (w/) MSE | FEDformer (w/) MAE | FEDformer (w/o) MSE | FEDformer (w/o) MAE |
> |---------|--------|--------------------|--------------------|---------------------|---------------------|
> | ETTm2   | 96     | 0.189              | 0.265              | 0.203               | 0.287               |
> |         | 192    | 0.250              | 0.311              | 0.269               | 0.328               |
> |         | 336    | 0.315              | 0.356              | 0.325               | 0.366               |
> |         | 720    | 0.415              | 0.405              | 0.421               | 0.415               |
> |         | Avg.   | 0.292              | 0.334              | 0.304               | 0.349               |
>
> **2**. We compared the performance of the NCDE corrector with that of MLP on the LTSF exchange dataset to demonstrate that it outperforms MLP. The MLP takes as input the forecasts from the Predictor and outputs errors of T-step forecasts directly. We used MLP as a direct multi-step predictor to model T-step errors, thereby avoiding error accumulation during autoregression.
>
> | Methods  | Metric | DLinear (w/ Neural CDE) MSE | DLinear (w/ Neural CDE) MAE | DLinear (w/ MLP) MSE | DLinear (w/ MLP) MAE | DLinear (w/o) MSE | DLinear (w/o) MAE |
> |----------|--------|------------------------------|------------------------------|-----------------------|-----------------------|--------------------|--------------------|
> | Exchange | 96     | **0.083**                        | 0.207                        | 0.084                 | **0.205**                 | 0.085              | 0.210              |
> | | 192    | **0.159**                        | **0.295**                        | **0.159**                 | 0.296                 | 0.162              | 0.297              |
> | | 336    | **0.243**                    | **0.370**                    | 0.301                 | 0.399                 | 0.333              | 0.442              |
> | | 720    | **0.482**                    | **0.548**                    | 0.692                 | 0.698                 | 0.896              | 0.724              |
> | | Avg.   | **0.242**                    | **0.355**                    | 0.309                 | 0.400                 | 0.369              | 0.418              |
>
> It is fair to ask for a comparison with other models that can model irregularly sampled time series. While ODE-RNN/GRU-ODE-Bayes and similar models can model irregularly sampled time series, they maintain a continuous hidden state between two observations, with a discrete jump at each observation. On the other hand, Neural CDE can maintain a continuous hidden state across observations. Our use case involves using Neural CDE to model error trajectories of forecasts from a Predictor. The forecasts from the Predictor are treated as observations by the Neural CDE. If the Predictor is continuous-time, e.g., NODE, it will generate continuous-time forecasts whose error trajectories also evolve continuously in time, making Neural CDE a natural fit for error modelling of forecasts from such Predictors. We talk about this in the second paragraph (Continuous-Time Models) of the related works section.
> Latent SDE is essentially a stochastic latent process that generates stochastic trajectories, unlike NCDE. We are not concerned with calibration or uncertainty quantification in the current work. Therefore, we focus on deterministic error modeling to mitigate forecast errors. To that end, NCDE is a model with a continuous-time latent state that evolves deterministically over time, and is a natural fit for continuous-time Predictors.

---

> ### Author Response · Authors · 2025-11-20
> **Comment [2/2]**
>
> **3**. We agree that the motivation is intuitive and lacks formal treatment. Our Predictor-Corrector framework works as follows:
> - The Predictor generates a full multi-step forecast trajectory $\{ \hat{x}_i \}\_{i=0}^{T-1}$
> - The Corrector takes this forecast as a control path, producing a continuous error trajectory $\{\hat{e}_i\}\_{i=0}^{T-1}$.
> - We form corrected forecasts $\hat{x}^{(c)}_i = \hat{x}_i + \hat{e}_i$.
>
> The proposed Corrector works only in the **forecast** regime where we don’t have access to observations. Therefore, we can’t augment the forecast with any additional observations. We only focus on deterministic Predictors in the current work, and we will include this assumption in the revised version. Hence, we don’t assume access to predictive variance. While our method should be able to correct the mean of a probabilistic Predictor that predicts mean and variance, we don’t make any claims in the paper about that.
> Regarding the **forecast error**, we **empirically** show that the MSE of corrected forecasts of Predictor over long horizons (400 timesteps) remains **well-bounded**. The results are shown for Lorenz, FHN, LVolt, and Glycolytic in the Figure (https://imgur.com/a/f7XmyIC). Each data point shows the log(MSE) from timestep 0 to timestep T on the x-axis.
>
> **4**. That is an interesting question. The Fig. 6 in the paper compares the predicted and corrected trajectories of Pen from the MuJoCo dataset. It can be observed that the corrected trajectories sometimes exacerbate the error in a few cases. We show two instances with green arrows in the Figure (https://imgur.com/a/7UhFKXi). In one case, the correction drives the predicted trajectory further away from ground truth. In the second case, it improves the performance. We believe that the cases where error amplifies are rare; otherwise, we wouldn’t have obtained a high percentage reduction in MSE of Contiformer on Pen in Table 2 with decent extrapolation performance.

---

### Official Review · Reviewer_EhtC · 2025-10-30

**Soundness:** 2
**Presentation:** 2
**Contribution:** 2
**Rating:** 2
**Confidence:** 4

**Summary:**

This paper proposes a Predictor-Corrector mechanism to address the problem of error accumulation in multi-step time-series forecasting. The core idea is to treat any pre-trained time-series model as a Predictor and use its output forecasts to drive a Corrector model. This Corrector is implemented as a Neural Controlled Differential Equation (Neural CDE).

The mechanism works as follows:
1. The Predictor generates a forecast trajectory.
2. This forecast trajectory is used as the continuous control path for the Neural CDE.
3. The Neural CDE is trained to predict the error (the residual between the Predictor's forecast and the ground truth).
4. The final, corrected forecast is the sum of the original prediction and the CDE's predicted error.

The authors demonstrate that this framework can be applied to diverse Predictors, including continuous-time (NODE, ContiFormer) and discrete-time (DLinear) models, and works on both regular and irregular data. The method is validated on synthetic, physics simulation, and real-world LTSF datasets, showing consistent performance improvements over the Predictor-alone baselines. The paper also introduces two regularization strategies, "Variable-Length Control Paths" and "Sparse Control Paths," to improve the Corrector's extrapolation capabilities.

**Strengths:**

1. The primary strength of this work is its agnosticism to the Predictor model. The ability to use the Corrector as a plug-and-play module to improve the performance of existing, pre-trained models—ranging from NODE to ContiFormer and even the non-continuous DLinear—is a significant practical advantage.
2. The paper provides a comprehensive set of experiments across synthetic data (Lorenz, FHN) , high-dimensional physics simulations (MuJoCo) , and challenging real-world LTSF benchmarks (Weather, Exchange, etc.). The consistent improvement in MSE/MAE across all these settings (e.g., in Table 1, 2, and 4) makes a strong case for the method's effectiveness.
3. The introduction of Variable-Length and Sparse Control Paths  is a clever and well-reasoned contribution. These regularizers directly address a known weakness of neural differential equations (extrapolation) and are shown to improve generalization while also accelerating training by reducing NFEs (as shown in Figure 4).

**Weaknesses:**

1. Over-reliance on the Decoder's Expressiveness: The entire framework hinges on the assumption that the CDE's latent state $z(t)$, controlled by the forecast path $X(s)$, will capture the necessary information to predict the error $e(t)$. However, the paper provides no structural or theoretical guarantee for this. The connection between $z(t)$ and $e(t)$ is established solely by a very powerful 4-layer MLP decoder ($FC(400)_4$). This design feels contrived. It is difficult to ascertain whether the CDE is truly learning the "error dynamics" or if it is merely acting as a complex feature extractor, with the heavy-lifting (i.e., fitting the complex $z$-to-$e$ mapping) being done by the high-capacity MLP decoder.

2. Limited Methodological Novelty: The core idea of using continuous-time neural models (NODEs/CDEs) to address shortcomings in time-series forecasting is not entirely unique. For instance, the provided paper by Jhin et al. (2024, "CONTIME") also employs a continuous-time (NODE-based GRU) architecture to specifically solve the problem of prediction delays in time-series models. While this paper's application (post-hoc error magnitude correction) is different from CONTIME's (end-to-end delay correction), the foundational concept of using CDEs/NODEs to fix a known flaw in time-series prediction is very similar. This concurrent work suggests that this methodological tool is a "next logical step" in the field, which somewhat diminishes the paper's claim to high conceptual novelty.

**Questions:**

1. Following Weakness #1: Could the authors provide an ablation study on the complexity of the decoder ($\xi_{\varphi}$)? Specifically, how does the Corrector's performance change if the 4-layer $FC(400)_4$ decoder is replaced with a simpler linear layer, or a 1-layer MLP? This would be crucial to disentangle the contribution of the CDE's dynamic modeling from the powerful function approximation capabilities of the decoder.
2. Following Weakness #2: Given that other concurrent work (e.g., Jhin et al., 2024) also uses continuous-time models (NODEs) to address related problems like prediction delay, how do the authors position the conceptual novelty of their contribution? Is the novelty in the idea of using CDEs for time-series, or purely in the application as a post-hoc correction module?

---

> ### Author Response · Authors · 2025-11-20
> **Comment [1/1]**
>
> **Weakness #1**: Thanks for a very insightful comment. We agree that an ablation study with a small decoder is crucial to demonstrate the expressiveness of the latent state $\mathbf{z}(t)$ of NCDE. The decoder ($\xi_{\varphi}$) for Neural CDE corrector maps the hidden state dynamics $\mathbf{z}(t)$ to error dynamics $\mathbf{\hat{e}}(t)$. Here, we show the results of FHN with varying decoder sizes. The results elsewhere are reported using a four-layer fully connected neural network with 400 neurons in each layer, denoted as $FC(400)_{4}$. With $FC(20)\_{1}$, we observe a performance degradation within the interpolation region with an improved extrapolation performance. This indicates that a small decoder mitigates overfitting and facilitates generalization, while the hidden state, $\mathbf{z}(t)$, effectively models error dynamics. The terms "Interp" and "Extra" stand for interpolation and extrapolation, respectively. The (%↓) shows a % reduction in MSE of the Predictor with the Corrector.
>
> | Decoder  | Model | Interp (20%) | Interp (50%) | Interp (80%) | Interp (100%) | Extra (20%) | Extra (50%) | Extra (80%) | Extra (100%) |
> |------------|-------|------------|------------|------------|-------------|-----------|-----------|-----------|------------|
> | $FC(20)\_1$  | w/o   | 0.225      | 0.161      | 0.150      | 0.137       | 0.264     | 0.183     | 0.200     | 0.180      |
> |  | w/    | 0.187      | 0.139      | 0.122      | 0.114       | 0.248     | 0.177     | 0.191     | 0.171      |
> |   | 0–t (%↓)   | 0–50 (17%) | 0–50 (14%) | 0–50 (18%) | 0–50 (17%)  | 0–90 (5%) | 0–190 (4%)| 0–220 (5%)| 0–200 (5%) |
> | $FC(100)\_1$ | w/o   | 0.225      | 0.161      | 0.150      | 0.137       | 0.274     | 0.178     | 0.192     | 0.179      |
> |  | w/    | 0.181      | 0.137      | 0.123      | 0.107       | 0.263     | 0.169     | 0.168     | 0.173      |
> |  | 0–t (%↓)  | 0–50 (18%) | 0–50 (12%) | 0–50 (18%) | 0–50 (22%)  | 0–100 (4%)| 0–150 (4%)| 0–200 (4%)| 0–180 (4%) |
> | $FC(400)\_4$ | w/o   | 0.225      | 0.161      | 0.150      | 0.137       | 0.242     | 0.178     | 0.192     | 0.166      |
> |  | w/    | 0.164      | 0.128      | 0.093      | 0.097       | 0.231     | 0.171     | 0.183     | 0.158      |
> |  | 0–t (%↓)  | 0–50 (27%) | 0–50 (20%) | 0–50 (38%) | 0–50 (29%)  | 0–75 (5%) | 0–150 (4%)| 0–140 (4%)| 0–150 (5%) |
>
> **Weakness #2**: As the reviewer has already pointed out, Jhin et al. (2024) addressed the prediction delay problem by using a continuous-time GRU, whereas our work employs NCDE as a post-hoc correction module. We believe that our contribution is orthogonal to the CONTIME. We claim novelty as follows:
> - We proposed a post-hoc continuous-time Corrector that works seamlessly for both continuous- and discrete-time Predictors. To the best of our knowledge, no other work introduced a corrector for continuous-time time-series models (NODE, Contiformer, etc.).
> - We introduced two regularization strategies to enhance the extrapolation performance of NCDE, and the reviewer has already acknowledged this in one of the comments. This contribution should also benefit future NCDE-based frameworks, demonstrating a broad impact of our proposed strategies beyond this work.
> - Our proposed Predictor-Corrector framework is agnostic to the Predictor, as demonstrated through a series of experiments. This demonstrates that our framework is applicable to diverse dynamic models (Predictors), and just not limited to time series forecasting tasks, unlike Jhin et al. (2024). To bolster this claim, we have added two new Predictors, e.g., RNN and TCN, in response to one of the reviews. These will be included in the revised version.

---

> > ### Comment · Reviewer_EhtC · 2025-11-24
> >
> > Thank you for the thorough response. The concerns I previously raised have now been fully clarified, and I appreciate the authors' efforts in strengthening the manuscript. I am satisfied with the revisions and will raise my score from 2 to 4.

---

> ### Author Response · Authors · 2025-12-01
>
> Thank you for revisiting our rebuttal. We appreciate your willingness to raise the score to 4 and the constructive feedback that helped strengthen the paper

---

### Official Review · Reviewer_jkpQ · 2025-10-30

**Soundness:** 2
**Presentation:** 3
**Contribution:** 2
**Rating:** 4
**Confidence:** 4

**Summary:**

The paper proposes a generic predictor–corrector scheme for time-series forecasting: a trained Predictor (e.g., NODE, ContiFormer, DLinear) produces a rollout, and a Neural CDE Corrector learns the residual trajectory and adds it back to improve forecasts. Research in Predictor corrector is important for many real world use cases. The authors claim the approach is model-agnostic (works for continuous- and discrete-time predictors, regular and irregular sampling) and introduce two “control-path” regularizers purported to improve extrapolation and reduce NFEs (function evaluations). Experiments span synthetic ODE systems, MuJoCo dynamics, and LTSF benchmarks; the paper reports consistent gains over the base predictors (with a few exceptions) and emphasizes irregular-sampling compatibility.

**Strengths:**

Clear, modular residual-learning formulation for error trajectories with a continuous-time model (Neural CDE) and a simple MSE loss; training/inference diagrams and algorithm are easy to follow.

Claimed agnosticism to predictor class and sampling regime is appealing for practitioners.

Order-agnostic correction. By modeling residual dynamics with a latent first-order CDE driven by a rich control path, the corrector can, in principle, compensate errors from higher-order systems (e.g., 2nd–4th order) without changing the predictor, provided sufficient history/derivative channels and capacity.

The  regularization ideas (variable-length / sparse control paths) presented seems intuitive and deployment friendly.

Broad empirical scope (synthetic, MuJoCo, and LTSF) demonstrates that the template can be dropped on several predictors.

**Weaknesses:**

Related Work / Positioning (major)

The paper omits central predictor–corrector lines that already cast learning as predict + update (observer/assimilation) or learned residual correction; directly overlapping the proposed idea. Some of the recent related works not cited/discusses/compared against:

PhyDNet (CVPR 2020): states that PhyCell is based on a prediction–correction paradigm inspired by data assimilation, disentangling a physics branch (predictor) from a residual branch (learned corrector) for video dynamics.

 Neural Koopman Prior for Data Assimilation (TSP 2024): a learned Koopman dynamical prior inside a variational data-assimilation loop; a learned predictor with correction via assimilation.

KalmanNet (IEEE TSP’22): NN-aided Kalman filtering that learns the gain/updates under partial/unknown dynamics; canonical “predict–update” with learning.

Semilinear Neural Operators: a unified recursive framework for prediction and data assimilation using observer design (ICLR 2024); explicitly combines prediction and correction operations for long-horizon PDEs with irregular/noisy measurements.

Predictor-Corrector Enhanced Transformers with Exponential Moving Average Coefficient Learning (Neurips 2024).

Related-Work paragraph on “Correctors in Forecasting” instead cites older ARIMA+ML hybrids and domain-specific error modeling, then claims novelty for a general corrector (Neural CDE) that “works for irregularly sampled time series.” This fails to acknowledge the modern ML-DA, observer, PhyDNet, and Koopman-assimilation literatures that already instantiate predictor–corrector at scale and in precisely the regimes you claim.

Conceptual & Technical gaps :

The choice to use forecasts as the sole control path is under-motivated. Why is this preferable to (i) augmenting with measurements/exogenous inputs, (ii) an observer-style gain (as in NODA), or (iii) a latent residual branch akin to PhyDNet? Provide analysis/theory for when a CDE-driven error process is identifiable/advantageous.

The paper asserts the corrector “works with irregularly sampled time series” but does not formalize stability or long-horizon behavior of the autoregressive correction (error accumulation remains a concern). A rigorous statement or bound (even empirical stability diagnostics) is missing.

The regularizers are framed as improving extrapolation and reducing NFEs, but this reads as empirical observation rather than a principle (no formal link between sparsity of the control path and generalization).

Weaknesses in experiments (Mostly Minor points)  :

Compute claims without measurements: You claim accelerated training from reduced NFEs; only qualitative NFE curves are shown. Provide wall-clock, solver tolerances, and NFEs for Predictor and Corrector (train & inference) against baselines.

Irregular sampling story is thin: While you claim compatibility with irregular sampling, the strongest empirical gains are on regular LTSF; the paper lacks strong irregular baselines (e.g., GRU-ODE-Bayes, ODE-RNN with assimilation updates) under identical missingness protocols.

Limited uncertainty reporting: Most tables lack std/CI across seeds; even the large LTSF table is point estimates only. ICLR standards call for robust seeds and uncertainty, especially with highly stochastic MuJoCo and long-horizon forecasts.

**Questions:**

Order-agnostic claim under-substantiated. The ability to correct higher-order dynamics hinges on observability (what signals the corrector sees), sampling rate, control-path engineering, and model capacity; the paper provides neither a formal analysis nor targeted experiments (e.g., explicit 4th-order regimes) to validate this claim.

How does a CDE-based corrector fundamentally differ from a more traditional observer-style update or kalman filter style correction? What advantages (expressivity, stability, identifiability) does control-path formulation have over these traditional approaches? Citing these works and discussing the contrasting advantage could enhance the positioning of the work.

From the looks of it the proposed method seems order agnostic for the underlying ODE. In my opinion a test that shows robustness/scalability to different order of ODEs will really help.

Evaluate against strong irregular baselines (e.g., ODE-RNN/GRU-ODE-Bayes) under matched missingness; does the CDE-corrector still win?

Do long-horizon rollouts exhibit bounded error under iterated correction? Provide diagnostics or a proposition delineating conditions for stability.

Since you correct forecasts, do you model predictive uncertainty or calibration? If not, how does the corrector affect coverage under conformal/quantile evaluation?

Ad-hoc training horizon choices: For LTSF you vary the corrector training length (50/100/…/336) and then use intermediate windows because very short/very long horizons hurt performance. This looks like post-hoc tuning to the sweet spot and should be replaced with a fixed, pre-registered protocol plus sensitivity/ablation.

In summary if the authors can improve their related work section adding discussions wrt both recent and traditional literature on different predictor corrector frameworks, can actually show how their corrector model behaves for different order of ODEs and observability it could significantly improve the overall contribution of their paper positively. In contrast, without these the paper claims seems overstated and lacks proper literature placement.

---

> ### Author Response · Authors · 2025-11-20
> **Comment [1/2]**
>
> - **Order of ODE and observability**: We select three systems of different orders (second, third, and fourth). The details of those systems are given in the Figure (https://imgur.com/a/X0Uvgkf). We train NODE as a Predictor for these examples with the same architectural details given in Appendix C.1. Both states and derivatives are modeled with NODE. The Corrector is trained with the same architectural details given in Appendix C.4. To simulate time-varying partial observability, we randomly mask half of the observed features for 0\%, 50\%, and 80\% of the observed time points. The results are shown in the Table. The interpolation (Interp) shows the performance of the Corrector for the first 50 timesteps, which corresponds to its training horizon. The extrapolation (Extra) columns lists the timesteps for every setting up to which the Corrector brings at least a 3\% reduction in MSE of NODE. The corrector consistently brings a reduction in the MSE of the Predictor irrespective of the levels of partial observability. However, the order of the ODE does impact the performance of the Corrector. The Corrector brings less than 10\% reduction in MSE within the interpolation region for the 4th order system compared to the 2nd and 3rd order systems, where the reduction in MSEs is significantly higher. Within the extrapolation region, the horizons up to which the Corrector brings at least 3\% reduction in MSE are much smaller for 4th order systems compared to 2nd and 3rd order systems. This demonstrates that the order of the system impacts the performance of the Corrector.  The terms "Interp" and "Extra" stand for interpolation and extrapolation, respectively. The (%↓) shows a % reduction in MSE of the Predictor with the Corrector.
>
> | Dynamical System | Model | Interp 0% | Interp 30% | Interp 60% | Extra 0% | Extra 30% | Extra 60% |
> |------------------|--------|-----------|-------------|-------------|-----------|------------|------------|
> | 2nd Order ODE    | w/o    | 0.0142    | 0.0452      | 0.0728      | 0.0413    | 0.0952     | 0.109      |
> |                  | w/     | 0.0039    | 0.0152      | 0.0252      | 0.0397    | 0.0922     | 0.097      |
> |                  | 0→t (%↓) | 0–50 (72%)  | 0–50 (66%)    | 0–50 (65%)    | 0–170 (4%)  | 0–180 (4%)   | 0–165 (10%)  |
> | 3rd Order ODE    | w/o    | 0.0101    | 0.0359      | 0.0934      | 7.3       | 8.1        | 9.4        |
> |                  | w/     | 0.0019    | 0.0039      | 0.0543      | 6.9       | 7.9        | 9.0        |
> |                  | 0→t (%↓) | 0–50 (81%)  | 0–50 (89%)    | 0–50 (41%)    | 0–180 (5%)  | 0–170 (4%)   | 0–190 (4%)   |
> | 4th Order ODE    | w/o    | 1.27      | 2.3         | 4.1         | 10.36     | 12.45      | 14.56      |
> |                  | w/     | 1.15      | 2.1         | 3.8         | 10.02     | 11.9       | 14.10      |
> |                  | 0→t (%↓) | 0–50 (9%)   | 0–50 (8%)     | 0–50 (7%)     | 0–90 (3%)   | 0–80 (4%)    | 0–100 (3%)   |
>
> - **CDE corrector and Filtering methods**. Kalman filters and classical observers are fundamentally measurement-driven state estimators. They require explicit process/measurement models, as well as online access to measurements. Our CDE-based Corrector instead operates in a forecast-only regime: at inference, it receives only the Predictor’s future forecast trajectory and models the dynamics of forecast errors. The errors are added to the forecast for correction. It does not attempt to estimate the state, hence classical notions of observability do not apply. The control-path formulation provides expressivity and flexibility that KF-style methods cannot offer: it naturally supports irregular sampling, continuous-time predictors (NODE, ContiFormer) and discrete-time predictors (DLinear), and enables control-path regularizations (sparse and variable-length) that substantially improve the extrapolation performance of the corrector.
> - **Baselines for irregular sampling**. While ODE-RNN/GRU-ODE-Bayes and similar models can model irregularly sampled time series, they maintain a continuous hidden state between two observations, with a discrete jump at each observation. On the other hand, Neural CDE can maintain a continuous hidden state across observations. Our use case involves using Neural CDE to model error trajectories of forecasts from a Predictor. The forecasts from the Predictor are treated as observations by the Neural CDE. If the Predictor is continuous-time, e.g., NODE, it will generate continuous-time forecasts whose error trajectories also evolve continuously in time, making Neural CDE a natural fit for error modelling of forecasts from such Predictors. We talk about this in the second paragraph (Continuous-Time Models) of the related works section.

---

> ### Author Response · Authors · 2025-11-20
> **Comment [2/2]**
>
> - **Predictive Uncertainty**. We don’t model predictive uncertainty or calibration. We assume a deterministic Predictor, and the corrector (Neural CDE) predicts a point estimate of error. Therefore, we don’t consider probabilistic evaluation metrics for this work.
> - **LTSF Horizon Choice**. We agree that we tuned the training horizon for LTSF datasets. The suggestion to train LTSF with intermediate horizon length is the same for all datasets and is supported by an ablation study in Table 11 in the Appendix. We recommend treating the training horizon as a hyperparameter and will mention this in the revised version. Hence, we don’t recommend a fixed, pre-registered protocol. The ablation study regarding the training horizon of LTSF datasets is given in Table 11 in the Appendix.
> - **Long-Horizon Rollouts**. We thank the reviewer for bringing the issue of long-horizon stability to our attention. We clarify first what “iterated correction” means in our setting, and then address empirical diagnostics. The Predictor-Corrector pipeline works as follows:
>    - The Predictor generates a full multi-step forecast trajectory $\{ \hat{x}_i \}\_{i=0}^{T-1}$
>    - The Corrector takes this forecast as a control path, producing a continuous error trajectory $\{\hat{e}_i\}\_{i=0}^{T-1}$.
>    - We form corrected forecasts $\hat{x}^{(c)}_i = \hat{x}_i + \hat{e}_i$.
>
>     There is no closed-loop where $\hat{x}^{(c)}$ is fed back into the Predictor or used as the new control path.
>      Regarding the forecast error, we empirically show that the MSE of corrected forecasts of Predictor over long horizons (400 timesteps) remains well-bounded. The results are shown for Lorenz, FHN, LVolt, and Glycolytic in the Figure (https://imgur.com/a/f7XmyIC). Each data point shows the log(MSE) from timestep 0 to timestep T on the x-axis.
>
> - **Positioning of our work in the literature**. We will add a paragraph in the related works section that includes recent works mentioned by the reviewer in their review. This will help us position our work compared to recent works on Predictor-Corrector frameworks in the related domains.

---

> > ### Comment · Reviewer_jkpQ · 2025-11-21
> >
> > Thank you for addressing my comments and clearing my doubts regarding the proposed work. Following the response from the authors I am more than happy to nudge the score towards acceptance once these changes reflect in the updated draft.

---

> ### Author Response · Authors · 2025-11-25
>
> Thank you for the thoughtful feedback and your intention to raise the score to 6. We have uploaded the revised manuscript. Please review the related works section and let us know if any additional changes are required.

---

### Official Review · Reviewer_w3E7 · 2025-11-01

**Soundness:** 3
**Presentation:** 3
**Contribution:** 3
**Rating:** 6
**Confidence:** 4

**Summary:**

This paper introduces a **Predictor–Corrector** framework for time-series forecasting in which any learned model (continuous- or discrete-time; e.g., NODE, ContiFormer, DLinear) acts as the **Predictor**, and a **Neural Controlled Differential Equation (Neural CDE)** serves as the **Corrector** that learns and adds back the Predictor’s **error dynamics**. The Corrector treats the Predictor’s multi-step forecast trajectory as a control path; a decoder maps the Neural CDE hidden state to residuals that are added to forecasts. Two regularizations aim to improve extrapolation and reduce training cost: **Variable-Length Control Paths** (randomly truncating the control path) and **Sparse Control Paths** (subsampling knots). Evaluations span **synthetic ODE systems** (Lorenz, Lotka–Volterra, FHN, Glycolytic), **MuJoCo physics** (Hopper, Walker2D, Pen, Hammer), and **LTSF** datasets (Exchange, ETTm2, ETTh2, ILI, Weather). Results show consistent MSE reductions versus base Predictors in interpolation and sustained improvements up to long horizons in several settings (e.g., DLinear gains on Exchange; ContiFormer gains on Walker2D/Pen). The study reports ablations on κ/η and the number of function evaluations (NFEs), indicating accuracy–extrapolation–efficiency trade-offs.

**Strengths:**

- **Originality.** This paper reframes residual correction as **continuous-time error-dynamics learning** using Neural CDEs driven by the Predictor’s trajectory, generalizing classic hybrid error-modeling beyond discrete, regularly sampled data.
- **Technical quality.** This paper presents a clear formulation (spline-based control paths, adjoint training, simple decoder) and systematic ablations of the κ/η regularizers, exposing accuracy–extrapolation–efficiency trade-offs (e.g., NFEs).
- **Clarity & reproducibility.** This paper clearly separates interpolation vs. extrapolation settings, provides algorithmic details and hyperparameter tables, and outlines data generation and training choices for replication.
- **Significance.** This paper shows consistent gains across heterogeneous Predictors (NODE, ContiFormer, DLinear) and domains (synthetic, physics, LTSF), indicating a practical, plug-in corrector for long-horizon forecasting—especially under irregular sampling.

**Weaknesses:**

- **Baselines for discrete tasks.** LTSF comparisons omit strong *residual* correctors (e.g., MLP/TCN residual heads, diffusion/consistency-style calibrators), making it unclear when a CDE-based corrector is strictly preferable.
- **Stability theory.** This paper lacks a formal analysis of stability/consistency for the closed-loop Predictor–Corrector under long-horizon rollouts; success criteria for extrapolation are largely empirical.
- **Efficiency reporting.** Beyond NFEs, this paper does not report wall-clock time, memory usage, or parameter counts versus simpler correctors, limiting claims about training/serving efficiency.
- **Predictor breadth.** Results are limited to NODE/ContiFormer/DLinear; adding RNN/TCN/graph forecasters would better substantiate the architecture-agnostic claim.
- **Interpretability.** This paper provides a limited qualitative analysis of learned corrections (e.g., whether they address phase drift, bias, or scale errors).

**Questions:**

1. **When is a CDE needed?** Please compare against residual MLP/TCN heads and diffusion/consistency-style correctors on LTSF to delineate regimes where the CDE brings clear advantages.
2. **Stability & robustness.** Provide long-horizon stress tests and sensitivity to interpolation schemes and ODE solvers; consider theoretical bounds or empirical diagnostics for divergence/overshoot.
3. **Training paradigm.** Compare two-stage vs. joint end-to-end (or alternating) training; discuss bias–variance–stability trade-offs and whether joint optimization improves extrapolation.
4. **Cost metrics.** Report runtime, memory, and parameter counts; include κ/η vs. accuracy **Pareto curves** to substantiate efficiency claims for the proposed regularizers.
5. **Predictor diversity & non-stationarity.** Evaluate on RNN/TCN/GRU-ODE-Bayes/graph predictors and regime-shifted splits; consider uncertainty/coverage metrics.
6. **Qualitative insight.** Visualize per-dimension corrections on real datasets to reveal which systematic Predictor biases are being fixed.

---

> ### Author Response · Authors · 2025-11-21
> **Comment [1/3]**
>
> - **When is CDE needed?** That is a valid question. However, our main motivation for a Neural CDE Corrector is not that “residual for every learned time-series model must be modeled with a CDE,” but that we want one unified Corrector architecture that (i) naturally handles irregularly sampled time series, (ii) can be paired with continuous-time Predictors (NODE, ContiFormer) and discrete-time Predictors (DLinear). On the LTSF benchmarks, the data are regularly sampled, and the Predictor (DLinear) is fully discrete. Here, the goal of our CDE Corrector is to **test generality**, not to claim that a CDE is strictly required. We agree that, in this purely discrete and regular regime, residual MLP/TCN heads and diffusion/consistency-style correctors are meaningful baselines. Therefore, we test diffusion and MLP on the Exchange dataset to demonstrate our competitive performance against these two models.
>
> | Dataset  | Horizon | DLinear (w/ Neural CDE) MSE | DLinear (w/ Neural CDE) MAE | DLinear (w/ MLP) MSE | DLinear (w/ MLP) MAE | DLinear (w/ Diffusion) MSE | DLinear (w/ Diffusion) MAE | DLinear (w/o) MSE | DLinear (w/o) MAE |
> |----------|--------|------------------------------|------------------------------|------------------------|------------------------|------------------------------|------------------------------|-------------------|-------------------|
> | Exchange | 96     | 0.083                        | 0.207                        | 0.084                 | 0.205                 | **0.071**                    | **0.195**                    | 0.085             | 0.210             |
> | | 192    | 0.159                        | 0.295                        | 0.159                 | 0.296                 | **0.152**                    | **0.284**                    | 0.162             | 0.297             |
> | | 336    | **0.243**                    | **0.370**                    | 0.301                 | 0.399                 | 0.301                        | 0.398                        | 0.333             | 0.442             |
> | | 720    | **0.482**                    | **0.548**                    | 0.692                 | 0.698                 | 0.576                        | 0.599                        | 0.896             | 0.724             |
> | | Avg.   | **0.242**                    | **0.355**                    | 0.309                 | 0.400                 | 0.275                        | 0.369                        | 0.369             | 0.418             |
>
> - **Long Horizon Stress Tests**. We empirically show that the MSE of corrected forecasts of Predictor over long horizons (400 timesteps) remains well-bounded. The results are shown for Lorenz, FHN, LVolt, and Glycolytic in the Figure (https://imgur.com/a/f7XmyIC). Each data point shows the log(MSE) from timestep 0 to timestep T on the x-axis.
>
> - **Two-stage vs. Alternate**. The Predictors and Correctors are trained in two different stages elsewhere in the paper. The two-stage training involves the Predictor first until convergence, and the Corrector learns the error dynamics of the Predictor afterwards. Another way is to train them jointly. A natural way of joint-training is to train them both alternatively, where Predictor and Corrector do gradient updates alternatively. The Predictor takes a few gradient steps and the Corrector does a few gradient updates to learn the updated error dynamics of Predictor. This essentially creates a moving target for the Corrector, making the training less stable compared to two-stage training. The alternate paradigm lowers the bias marginally, which manifests itself in the form of performance degradation within interpolation. Empirically, we did not observe any improvement in extrapolation with alternate training and believe that two-stage training achieves a better bias-variance-stability trade-off.
>
> | Dynamical System | Model | Interp Two-stage | Interp Alternate | Extra Two-stage | Extra Alternate |
> |------------------|--------|------------------|-------------------|------------------|-------------------|
> | Lorenz           | w/o    | 0.887            | 0.870             | 6.464            | 4.102             |
> |                  | w/     | 0.468            | 0.625             | 6.245            | 3.901             |
> |                  | 0→t %↓ | 0–50 47%         | 0–50 28%          | 0–150 3%         | 0–120 4%          |
>
> - **Per-dimension correction**. We show the per-dimension corrections on the Exchange dataset in the Figure (https://imgur.com/a/r8KJhOp). It can be seen that the Predictor underestimates consistently on dimensions (Dim.) 4,7, \&8, and the Corrector systematically corrects it by bringing the predicted trajectory closer to the ground truth. The Predictor shows decent performance on Dim. 1, and the Corrector leaves it untouched. This indicates that the Corrector learns dimension-specific error dynamics rather than applying a uniform correction to each dimension.

---

> ### Author Response · Authors · 2025-11-21
> **Comment [2/3]**
>
> - **Sensitivity to interpolation schemes**.The $\mathtt{diffrax}$ library contains different interpolation schemes for control paths of Neural CDE. Here, we present the sensitivity of Corrector performance to those interpolation schemes. The results with Cubic interpolation for Walker2D and Hammer are copied from Table 2 in the paper. The Linear interpolation brings a slightly smaller reduction in MSE (\%) compared to the Cubic interpolation in almost every setting in the Table below. This observation is in line with the results reported by [1]. The terms "Interp" and "Extra" stand for interpolation and extrapolation, respectively. The (%↓) shows a % reduction in MSE of the Predictor with the Corrector.
>
> | System               | Model | Interp 20% | Interp 50% | Interp 80% | Interp 100% | Extra 20% | Extra 50% | Extra 80% | Extra 100% |
> |----------------------|--------|------------|------------|------------|-------------|-----------|-----------|-----------|------------|
> | Walker2D (Linear)    | w/o    | 1.826      | 0.572      | 0.408      | 0.164       | 2.72      | 0.577     | 0.778     | 0.869      |
> |                      | w/     | 1.029      | 0.435      | 0.228      | 0.073       | 2.62      | 0.532     | 0.758     | 0.839      |
> |                      | 0→t (%↓) | 0–50 43%   | 0–50 23%   | 0–50 43%   | 0–50 55%    | 0–180 4%  | 0–100 7%  | 0–130 3%  | 0–140 3%   |
> | Walker2D (Cubic)     | w/o    | 1.826      | 0.572      | 0.408      | 0.164       | 2.89      | 1.70      | 0.778     | 0.869      |
> |                      | w/     | 0.721      | 0.285      | 0.149      | 0.053       | 2.68      | 1.63      | 0.750     | 0.833      |
> |                      | 0→t (%↓) | 0–50 61%   | 0–50 50%   | 0–50 63%   | 0–50 68%    | 0–190 7%  | 0–186 4%  | 0–130 4%  | 0–140 4%   |
> | Hammer (Linear)      | w/o    | 0.020      | 0.0137     | 0.0106     | 0.0085      | 0.0115    | 0.0083    | 0.0049    | 0.0046     |
> |                      | w/     | 0.012      | 0.0058     | 0.0038     | 0.0039      | 0.0110    | 0.0080    | 0.0047    | 0.0045     |
> |                      | 0→t (%↓) | 0–50 39%   | 0–50 57%   | 0–50 63%   | 0–50 54%    | 0–140 4%  | 0–140 3%  | 0–180 4%  | 0–150 6%   |
> | Hammer (Cubic)       | w/o    | 0.020      | 0.0137     | 0.0106     | 0.0085      | 0.0158    | 0.0073    | 0.0049    | 0.0046     |
> |                      | w/     | 0.012      | 0.0059     | 0.0037     | 0.0030      | 0.0150    | 0.0070    | 0.0047    | 0.0044     |
> |                      | 0→t (%↓) | 0–50 37%   | 0–50 56%   | 0–50 64%   | 0–50 63%    | 0–100 4%  | 0–150 4%  | 0–180 6%  | 0–150 6%   |
>
> - **Predictor Diversity**. To bolster our claim that the proposed Predictor-Corrector framework is agnostic to the underlying Predictor, we evaluate our Corrector on two additional Predictors, i.e., RNN and TCN. The datasets are Lorenz and FHN. The extrapolation columns list the horizon up to which the Corrector brings at least 3\% reduction in MSE of Predictor.
>
> | Dynamical System      | Model | Interp RNN | Interp TCN | Extra RNN | Extra TCN |
> |------------------------|--------|------------|------------|-----------|-----------|
> | Lorenz                 | w/o    | 1.102      | 0.725      | 5.705     | 3.908     |
> |                        | w/     | 0.901      | 0.635      | 5.504     | 3.709     |
> |                        | 0→t %↓ | 0–50 18%   | 0–50 13%   | 0–80 3%   | 0–100 5%  |
> | Glycolytic Oscillator  | w/o    | 0.190      | 0.140      | 0.187     | 0.160     |
> |                        | w/     | 0.140      | 0.110      | 0.180     | 0.150     |
> |                        | 0→t %↓ | 0–50 26%   | 0–50 21%   | 0–90 4%   | 0–110 6%  |
>
>
> [1]. James Morrill, Patrick Kidger, Lingyi Yang, and Terry Lyons. On the choice of interpolation scheme for neural cdes. Transactions on Machine Learning Research, 2022(9), 2022.

---

> ### Author Response · Authors · 2025-11-21
> **Comment [3/3]**
>
> **Sensitivity to ODE solvers**. The $\mathtt{diffrax}$ package has different explicit Runga-Kutta (RK) methods. We used $\mathtt{Tsit5}$ to report results in the paper everywhere else. Here, we analyze the sensitivity of Walker2D results to other solvers from the explicit RK family. The results are reported in Table below. The $\mathtt{Tsit5}$ performed better both in interpolation and extrapolation regions compared to other solvers, followed by $\mathtt{Dopri5}$. The adaptive-step size solvers, such as $\mathtt{Heun}$, $\mathtt{Dopri5}$, \& $\mathtt{Tsit5}$, performed better than $\mathtt{Euler}$ method.
>
> | ODE Solver | Model | Interp 20% | Interp 50% | Interp 80% | Interp 100% | Extra 20% | Extra 50% | Extra 80% | Extra 100% |
> |------------|--------|-------------|-------------|-------------|--------------|------------|------------|------------|-------------|
> | Euler      | w/o    | 1.826       | 0.572       | 0.408       | 0.164        | 2.72       | 0.639      | 0.778      | 0.245       |
> |            | w/     | 1.375       | 0.437       | 0.292       | 0.105        | 2.62       | 0.620      | 0.743      | 0.234       |
> |            | 0→t %↓ | 0–50 24%    | 0–50 23%    | 0–50 28%    | 0–50 36%     | 0–180 4%   | 0–100 3%   | 0–130 4%   | 0–80 4%     |
> | Heun       | w/o    | 1.826       | 0.572       | 0.408       | 0.164        | 1.86       | 0.969      | 0.778      | 0.413       |
> |            | w/     | 0.901       | 0.359       | 0.328       | 0.078        | 1.67       | 0.925      | 0.752      | 0.394       |
> |            | 0→t %↓ | 0–50 50%    | 0–50 37%    | 0–50 19%    | 0–50 52%     | 0–130 9%   | 0–130 4%   | 0–130 4%   | 0–120 4%    |
> | Dopri5     | w/o    | 1.826       | 0.572       | 0.408       | 0.164        | 1.86       | 0.969      | 0.778      | 0.869       |
> |            | w/     | 0.819       | 0.361       | 0.329       | 0.064        | 1.81       | 0.929      | 0.760      | 0.841       |
> |            | 0→t %↓ | 0–50 55%    | 0–50 36%    | 0–50 19%    | 0–50 60%     | 0–130 3%   | 0–130 4%   | 0–130 3%   | 0–140 4%    |
> | Tsit5      | w/o    | 1.826       | 0.572       | 0.408       | 0.164        | 2.89       | 1.70       | 0.778      | 0.869       |
> |            | w/     | 0.721       | 0.285       | 0.149       | 0.053        | 2.68       | 1.63       | 0.750      | 0.838       |
> |            | 0→t %↓ | 0–50 61%    | 0–50 50%    | 0–50 63%    | 0–50 68%     | 0–190 7%   | 0–180 4%   | 0–130 4%   | 0–140 4%    |
>
> - **Pareto Curves**. The $\kappa$ (Sparse Control Paths) and $\eta$ (Variable Length Control Paths) regularization strategies are two proposed approaches to enhance the extrapolation performance and computational efficiency of the NCDE corrector. Both approaches offer a trade-off between the number of function evaluations (NFEs) and the extrapolation horizon. The extrapolation horizon is the timestep up to which the corrector brings at least 3\% reduction in MSE. We plot Pareto curves for both $\kappa$ and $\eta$ to show the trade-off between NFE and extrapolation horizon (https://imgur.com/a/IU66U7R). Pareto curves for $\kappa$ \& $\eta$ show the value of $\kappa$ \& $\eta$ right next to each data point, while the x and y axes show the NFE and extrapolation horizon, respectively. It can be observed that achieving both a small NFE and a large extrapolation horizon is challenging. The best point is the one that balances both NFE and the extrapolation horizon.
>
> - **Wall-Clock time**. We report the average wall-clock time it takes to complete one epoch during training, varying the levels of $\kappa$ and $\eta$, in the Figure (https://imgur.com/a/geaKrBW). It can be seen that smaller values of $\kappa$ and larger values of $\eta$ result in faster training, corroborating the results shown in Figure 4 of the paper.

---

### Author Response · Authors · 2025-11-25
**Revised Manuscript and Rebuttal Highlights**

We thank all the reviewers for their constructive feedback, which helped us significantly strengthen our manuscript. We highlight the changes made to the manuscript during the revision process. The following new sections have been added.
- Revised Related Works section
- Appendix G.2 - Order of ODE
- Appendix G.3 - Sensitivity to interpolation schemes
- Appendix G.4 - Sensitivity to ODE solvers
- Appendix G.5 - Decoder Size
- Appendix G.6 - Efficiency-Extrapolation Pareto Curves for kappa and eta regularization
- Appendix G.7 - Two-Stage versus Alternate Training
- Appendix G.8 - Wall-clock time of kappa and eta
- Appendix G.9 - Long-Horizon Stress Tests
- Appendix H.1 contains additional LTSF results where we improve the performance of DLinear with Neural CDE (ours),  MLP, and Diffusion correctors for comparison on the Exchange dataset.
- Appendix H.2 contains results on improvements in the performance of TCN and RNN with our corrector.

We summarize a few important and common responses from the rebuttal.
- **Long-Horizon Tests**: A few reviewers raised concerns about the well-boundedness of the error of the corrected forecasts of the Predictors. We conduct stress tests to show well-boundedness.
- **Positioning of our work**: We revise the related works section with recent works on Predictor-Corrector in mind to better position our work within the existing literature.
- **Baselines**: A few reviewers raised concerns about comparisons to ODE-RNN and other related continuous-time models. We addressed these issues by highlighting the limitations of other models and explaining how Neural CDE provides a solution.
- **Observations**: A few reviewers raised concerns about not taking observations into account while correcting forecasts. We addressed this issue by emphasizing the fact that our Corrector works in a forecast regime where we don’t have access to observations. Therefore, the Corrector relies solely on the Predictor’s forecasts for correction.
- **CDE for LTSF**: It is reasonable to question the need for CDE for discrete, regular sampled time-series tasks. We emphasize that our motivation for testing Neural CDE for LTSF was to assess its generality and propose it as a unified corrector for both continuous and discrete, as well as regularly and irregularly sampled tasks.

We encourage all **reviewers** to **review** the **revised manuscript** and let us know if any further changes are required.

---

### Meta-Review · Area_Chair_ffeG · 2025-12-19

**Summary:**

This submission targets on the problem of the time-series problem through neural CDE to correct error dynamics. Reviews raised questions about theoritical analysis, motivation of proposed method (especially the controlled path used and over-reliance of decoder), and experimental comparison.

**Reviewer Concerns:**

The AC has carefully read though the submission, reviewers' comments and rebutal. While AC agrees that the concerns regarding experimental evaluations have been properly addressed, the rebuttal of theoritical analysis, technical novelty (two reviewers raised comments regarding the comparison of exisitng works) are still limited.

**Reviewer Scores:**

Some reviewers already provided feedback after rebuttal period, while one reviewer agreed to change the score to acceptance, another reviewer still had concerns (changed the score from 2 to 4). The AC believes the remaining reviewer is less likely to upgrade the score to acceptance, since the authors also acknowledged the limitation (raised by the reviewer) of the proposed method.

---

### Decision · Program_Chairs · 2026-01-26

Reject